# DOME: Taming Diffusion Model into High-Fidelity Controllable Occupancy World Model

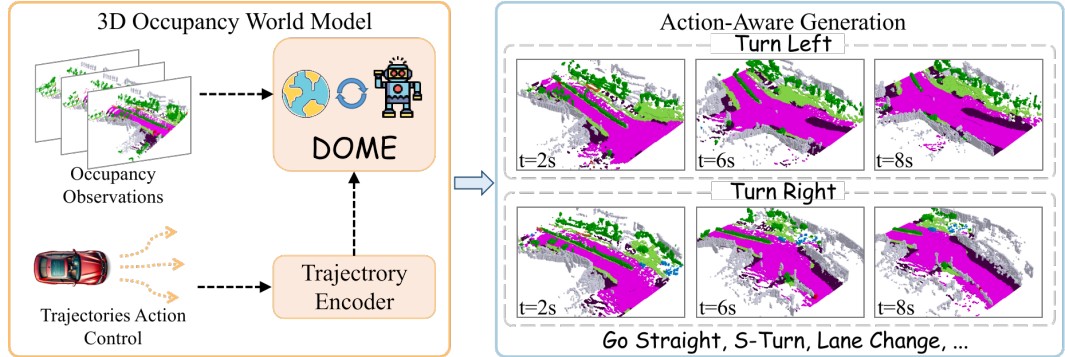

Figure 1: **Our Occupancy World Model** can generate long-duration occupancy forecasts and can be effectively controlled by trajectory conditions.

## Abstract

We propose **DOME**, a diffusion-based world model that predicts future occupancy frames based on past occupancy observations. The ability of this world model to capture the evolution of the environment is crucial for planning in autonomous driving. Compared to 2D video-based world models, the occupancy world model utilizes a native 3D representation, which features easily obtainable annotations and is modality-agnostic. This flexibility has the potential to facilitate the development of more advanced world models. Existing occupancy world models either suffer from detail loss due to discrete tokenization or rely on simplistic diffusion architectures, leading to inefficiencies and difficulties in predicting future occupancy with controllability. Our DOME exhibits two key features: (1) **High-Fidelity and Long-Duration Generation**. We adopt a spatial-temporal diffusion transformer to predict future occupancy frames based on historical context. This architecture efficiently captures spatial-temporal information, enabling high-fidelity details and the ability to generate predictions over long durations. (2) **Fine-grained Controllability**. We address the challenge of controllability in predictions by introducing a trajectory resampling method, which significantly enhances the model's ability to generate controlled predictions. Extensive experiments on the widely used nuScenes dataset demonstrate that our method surpasses existing baselines in both qualitative and quantitative evaluations, establishing a new **state-of-the-art** performance on nuScenes. Specifically, our approach surpasses the baseline by 10.5% in mIoU and 21.2% in IoU for occupancy reconstruction and by 36.0% in mIoU and 24.6% in IoU for 4D occupancy forecasting.

## 1 Introduction

Autonomous driving has recently benefited from rapidly advancing learning techniques and increasingly sophisticated data collection pipelines (Chen et al., 2024). However, significant challenges remain, such as the long-tail distribution and corner cases, which are difficult to address even with the state-of-the-art (SOTA) methods (Hu et al., 2023b) or extensive data collection efforts. A promising approach to addressing these challenges lies in world models. World models incorporate historical

context and alternative agents' actions to predict the future evolution of environmental observations. This allows the autonomous driving model to anticipate further into the future, improving the evaluation of action viability (Yang et al., 2023).

World models can be categorized into several types, including 2D video-based models and 3D representation-based models, such as those utilizing LiDAR and occupancy frameworks. While video-based world models have shown considerable success in predicting realistic camera observations, they still face challenges in maintaining cross-view and cross-time consistency. These limitations hinder their applicability in real-world scenarios. On the other hand, recent occupancy-based world models naturally avoid this issue. These models take historical occupancy sequences as input and predict future occupancy observations, benefiting from the **raw 3D representation** that ensures intrinsic 3D consistency. Moreover, occupancy annotations are relatively easy to acquire, as they can be efficiently learned from sparse LiDAR annotations (Tian et al., 2023) or potentially through self-supervision from temporal frames. Occupancy-based models are also modality-agnostic, meaning that they can be generated from monocular or surrounding cameras (Zheng et al., 2024), or from LiDAR sensors (Zuo et al., 2023).

Existing occupancy world models can be categorized into two types: autoregressive-based and diffusion-based. Autoregressive-based methods (Zheng et al., 2023; Wei et al., 2024) predict future occupancy using discrete tokens in an autoregressive manner. However, because these methods rely on discrete tokenizers, the process of quantization results in information loss, which limits the ability to predict high-fidelity occupancy. Moreover, autoregressive methods struggle to generate realistic long-duration occupancy sequences because training GPT-based methods is challenging. Diffusion-based approach (Wang et al., 2024) flattens spatial and temporal information into a one-dimensional sequence of tokens rather than separating and processing them individually, causing struggles to capture spatial-temporal information efficiently. Consequently, integrating historical occupancy information into the model becomes difficult because spatial and temporal data are combined. This limitation means the model can generate outputs but cannot predict, restricting its applicability in real-world scenarios. Furthermore, we found that most occupancy world models demonstrate insufficient exploration of fine-grained control, leading to overfitting to specific scenes and limiting their applicability to downstream tasks.

To address the aforementioned issues, we propose a novel method for predicting future occupancy frames, called **DOME**. Specifically, our approach consists of two components: the Occ-VAE and the spatial-temporal diffusion transformer. To overcome the limitations of discrete tokens, our Occ-VAE utilizes a continuous latent space to compress occupancy data. This allows for effective compression while preserving high-fidelity details. Our world model demonstrates two key features: (1) **High-Fidelity and Long-Duration Generation**. We employ a spatial-temporal diffusion transformer to predict future occupancy frames. By utilizing contextual occupancy conditioning, we incorporate historical occupancy information as input. The spatial-temporal architecture efficiently captures both spatial and temporal information, resulting in fine details and enabling the generation of long-duration predictions (32s). (2) **Fine-grained Controllability**. We address the challenge of precise control with trajectories, particularly the issue that occupancy predictions often fail to accurately capture the diverse actions of the ego vehicle. To enhance controllability, we propose a trajectory resampling method, which significantly improves the model's ability to generate more precise and varied occupancy predictions. We conducted experiments on the widely used nuScenes benchmark (Caesar et al., 2019), and the quantitative results demonstrate that our method can achieve SOTA performance in both 3D occupancy reconstruction and 4D occupancy prediction. Our approach outperforms the baseline by a significant margin, with a 36.0% improvement in mIoU and a 24.6% improvement in IoU.

To summarize, our contributions are as follows:

- We propose **DOME**, a novel diffusion-based world model that predicts future occupancy frames based on historical occupancy observations. It incorporates Occ-VAE, which utilizes a continuous latent space for high-fidelity occupancy compression, and a spatial-temporal diffusion transformer for efficient 4D occupancy prediction.

- We address the challenge of precise control using trajectory conditions, introducing a trajectory resampling method to enhance controllability, which significantly improves the control capabilities of our world model.

- Experimental results demonstrate that our method achieves SOTA performance on the nuScenes dataset for both 3D occupancy reconstruction and 4D occupancy prediction.

## 2 RELATED WORK

### 2.1 3D OCCUPANCY PREDICTION

The task of 3D occupancy prediction involves predicting both the occupancy status and the semantic label of each 3D voxel (Zhang et al., 2023; Huang et al., 2023; Li et al., 2023b). Recent approaches (Huang et al., 2023; Li et al., 2023b) have focused on vision-based occupancy prediction, utilizing images as input. These methods can be categorized into three mainstream types based on their feature enhancement: Bird's Eye View (BEV), Tri-Perspective View (TPV), and voxel-based methods.

The BEV-based method (Li et al., 2023b; Philion & Fidler, 2020) learns features in BEV space, which is less sensitive to occlusion. It first extracts 2D image features using a backbone network, applies a viewpoint transformation to obtain BEV features, and finally uses a 3D occupancy head for prediction. However, BEV methods struggle to convey detailed 3D information due to their top-down projection. To address this limitation, TPV-based methods (Huang et al., 2023; Zuo et al., 2023) leverage three orthogonal projection planes, enhancing the ability to describe fine-grained 3D structures. These methods also extract 2D image features, which are then lifted to three planes before summing the projected features to form the 3D space representation. In contrast to these projection-based approaches, voxel-based methods (Li et al., 2023a; Zheng et al., 2024) directly learn from the raw 3D space, effectively capturing comprehensive spatial information. These methods extract 2D image features from a backbone network and transform them into a 3D representation, which is subsequently processed by a 3D occupancy head to make occupancy predictions.

### 2.2 AUTONOMOUS DRIVING WORLD MODEL

The world model is a representation of the surrounding environment of an agent (Ha & Schmidhuber, 2018). Given the agent's actions and historical observations, it predicts the next observation, helping the agent develop a comprehensive understanding of its environment. The most popular approach involves predicting images or videos of driving scenes (Hu et al., 2023a; Zhao et al., 2024; Su et al., 2024). These methods can be considered as driving simulators, as they generate front-view or range-view outputs from car cameras. Hu et al. (2023a) introduces GAIA-1, a generative world model for autonomous driving that uses video, text, and action inputs to create realistic driving scenarios.

Recent methods aim to extend the autonomous driving world model by incorporating different modalities, such as point clouds (Zhang et al., 2024; Zyrianov et al., 2024), or 3D occupancy (Ma et al., 2023; Wang et al., 2024). LiDAR-based world models forecast 4D LiDAR point clouds. Zhang et al. (2024) propose Copilot4D, a world modeling approach using VQVAE and discrete diffusion to predict future observations. It improves prediction accuracy by over 50% on several datasets, showcasing the potential of GPT-like unsupervised learning in robotics. Another approach is the occupancy-based world model, which forecasts future scenes via 3D occupancy. Zheng et al. (2023) introduce OccWorld, a 3D world model for autonomous driving that predicts ego car movement and surrounding scene evolution using 3D occupancy. Wang et al. (2024) propose OccSora, a diffusion-based model for simulating 3D world development in autonomous driving. It uses a 4D scene tokenizer and a DiT world model for occupancy generation, aiding decision-making in autonomous driving. However, it focuses solely on generating occupancy rather than predicting observations based on historical data, raising questions about its efficacy as a world model and limiting its applicability in realistic scenarios.

## 3 METHOD

In this section, we introduce **DOME**, a diffusion-based occupancy world model. Our method consists of two main components: Occ-VAE Sec. 3.1 and DOME Sec. 3.2. To align the world model with trajectory conditions, we present a trajectory encoder and a trajectory resampling technique, specifically designed to enhance the model's controllability, as described in Sec. 3.3. Finally, we demonstrate the applications of our DOME in Sec. 3.4.

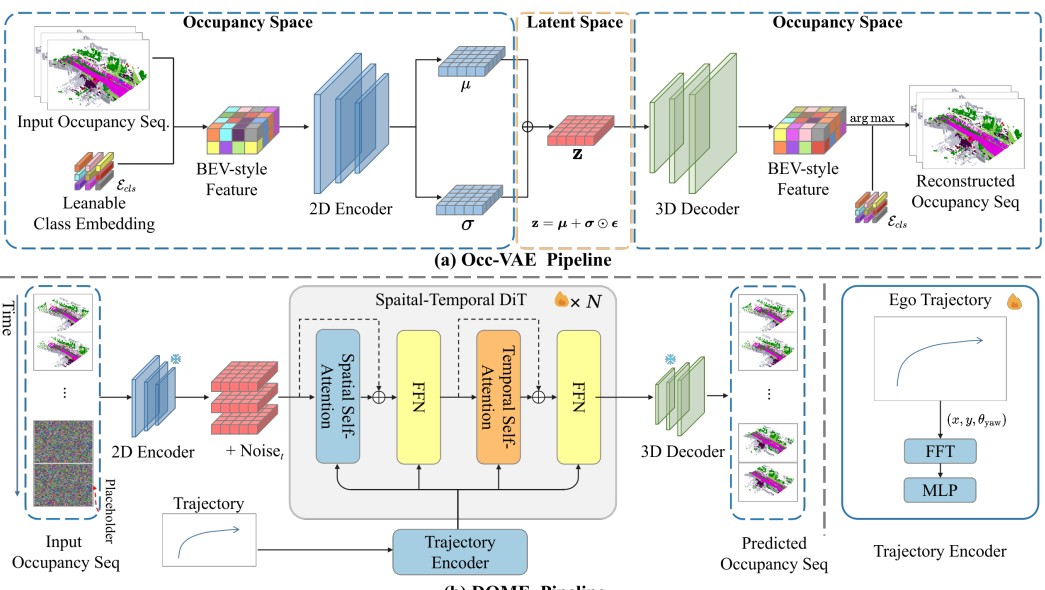

Figure 2: **(a): Occ-VAE Pipeline.** This component encodes occupancy frames into a continuous latent space, enabling efficient data compression. **(b): DOME Pipeline.** This component learns to predict 4D occupancy based on historical occupancy observations.

### 3.1 OCC-VAE

Occ-VAE is a core component of our model, utilizing a variational autoencoder (VAE) (Kingma & Welling, 2013) to compress occupancy data into a latent space, which is essential for improving the representation compactness and the efficiency of world model predictions. Noticing that discrete tokenizers often fail to retain the fine details of occupancy frames, we propose encoding the dense occupancy data into a continuous latent space to better preserve intricate spatial information. The proposed architecture, as illustrated in Fig. 2, is detailed as follows:

**Occupancy Data:** As Occ-VAE is specifically designed for occupancy data, we begin by discussing this 3D scene representation. The 3D occupancy data $\mathbf{x} \in \mathbb{R}^{H \times W \times D}$ voxelizes the surrounding environment of the ego vehicle into an $H \times W \times D$ voxel grid, where each grid cell is assigned semantic labels based on the objects it contains.

**Encoder**: Inspired by image-based VAE methods (Kingma & Welling, 2013), we propose a continuous VAE specifically designed for occupancy data. To handle the 3D occupancy data $\mathbf{x}$, which consists of discrete semantic IDs, we first transform it into a Bird's Eye View (BEV) style tensor $\mathbf{x}_{bev} \in \mathbb{R}^{H \times W \times DC_{emb}}$ by indexing a learnable class embedding $\mathcal{E}_{cls} \in \mathbb{R}^{n \times C_{emb}}$. This process flattens the occupancy data into a consistent feature dimension. Subsequently, an encoder network $q_\phi(\mathbf{z} \mid \mathbf{x})$ encodes the transformed data into a compressed representation. This representation is then split into $\boldsymbol{\mu} \in \mathbb{R}^{n_h \times n_w \times C}$ and $\boldsymbol{\sigma} \in \mathbb{R}^{n_h \times n_w \times C}$ along the channel dimension, where $n_h$ and $n_w$ represent the spatial dimensions of the encoded data, and $C$ denotes the channel dimension. After encoding, the continuous latent variable $\mathbf{z} \sim q_\phi(\mathbf{z} \mid \mathbf{x})$ is sampled using the reparameterization trick, following the approach used in image-based VAEs (Kingma & Welling, 2013): $\mathbf{z} = \boldsymbol{\mu} + \boldsymbol{\sigma} \odot \boldsymbol{\epsilon}$, where $\boldsymbol{\epsilon} \sim \mathcal{N}(0, \boldsymbol{I})$ is a noise vector sampled from a standard normal distribution, and $\odot$ denotes element-wise multiplication.

The encoder incorporates both 2D convolutional layers and attention blocks. The class embedding $\mathcal{E}_{cls}$ is initialized randomly and trained jointly with the Occ-VAE.

**Decoder**: The decoder network $p_\theta(\mathbf{x} \mid \mathbf{z})$ is responsible for reconstructing the input occupancy from the sampled latent variable $\mathbf{z}$. It employs 3D deconvolution layers to upsample the latent representation, ensuring improved temporal consistency (Blattmann et al., 2023). The upsampled features $\mathcal{F}$ are then reshaped into $H \times W \times D \times C_{emb}$. The logits score $s$ is computed through the dot product with the class embedding, where the $\arg\max$ of the logits determines the final class prediction.

**Training Loss**: During the training of Occ-VAE, our loss function consists of two components: the reconstruction loss and the KL divergence loss, following the standard VAE framework (Kingma & Welling, 2013). We employ cross-entropy loss as the reconstruction loss. Additionally, to address class imbalance in predictions, we incorporate the additional Lovasz-softmax loss following (Berman et al., 2018), which helps alleviate the imbalance issue. The total loss is defined as follows:

$$L_{\text{Occ-VAE}} = \mathcal{L}_{CE}(\mathbf{x}, s) + \beta D_{KL}\left(q_\phi(\mathbf{z} \mid \mathbf{x}) \| p(\mathbf{z})\right) + \lambda L_{lovasz}(\mathbf{x}, s) \quad (1)$$

where $\lambda$ and $\beta$ are the loss weights for the Lovasz-softmax loss and the KL divergence loss, respectively. After training, the Occ-VAE model is frozen, with its encoder serving as a feature extractor to obtain latent representations for DOME training, while its decoder reconstructs the latent representations from DOME to generate occupancy data.

## 3.2 DOME: A Diffusion-based Occupancy World Model

Occupancy world models predict future occupancy observations $o_t$ based on the agent's historical data $(o_1, a_1, \ldots, o_{t-1}, a_{t-1})$, where $o$ represents occupancy observations and $a$ denotes the agent's actions. To achieve this, we employ a latent diffusion model with temporal-aware layers, which enables the model to effectively learn from temporal variations. Historical occupancy observations are integrated using a temporal mask, encouraging the model to learn to predict future frames based on the conditional frame. Furthermore, to provide the world model with enhanced motion priors and controllability, our trajectory encoder incorporates the ego vehicle's actions, allowing for precise next-frame predictions controlled by given camera poses. Specifically, our model takes as input an encoded latent $\mathbf{z} \in \mathbb{R}^{n_f \times n_h \times n_w \times C}$ along with the ego vehicle's trajectory as input, where $n_f$ represents the temporal dimension corresponding to the number of frames in the 4D occupancy data. The latent is partially masked, allowing visibility for only $n_c$ frames ($n_c < n_f$), and the model is trained to predict the remaining masked frames.

**Spatial-Temporal Diffusion Transformer**: To predict future occupancy with temporal awareness, we adopt a spatial-temporal latent diffusion transformer inspired by video-based methods (Ma et al., 2024). We first patchify the latent representation $\mathbf{z}$ into $n_f$ frames of sequence tokens, with each sequence containing $n_t = \frac{n_h}{p} \times \frac{n_w}{p}$ tokens, where $p$ represents the patch size. Positional embeddings are then added to both the spatial and temporal dimensions (see the appendix for details). As illustrated in Fig. 2, our model is composed of two fundamental types of blocks: spatial blocks and temporal blocks. The spatial blocks capture spatial information across frames that share the same temporal index, while the temporal blocks extract temporal information along the temporal axis at a fixed spatial index. These blocks are arranged in a staggered fashion to effectively capture both spatial and temporal dependencies, as shown in Fig. 2.

**Historic Occupancy Condition**: To enable the model to predict future occupancy features, it is essential to condition the generation on historical occupancy data. This is achieved using a conditioning mask. Given a multi-frame context of occupancy data and a hyperparameter $n_c$ representing the number of context frames, the latent $z_c$ is encoded from the historical occupancy observations. We then construct a conditioning mask $\mathcal{M} = [t < n_c \mid t \in \{0, 1, 2, \ldots, n_f\}]$, which ensures that the model conditions its predictions on the available context frames. During training, the noised tokens $z_i$ are partially replaced by the context latents according to the condition mask for any training iteration that uses context frames:

$$\hat{z}_i = \mathcal{M} \cdot z_c + (1 - \mathcal{M}) \cdot z_i. \quad (2)$$

To enable the model to generate without conditioning, we apply a dropout mechanism in which, for a fixed proportion of iterations, the model is trained without context frames.

**Loss Function**: We extend the vanilla diffusion loss to a spatial-temporal version, making it compatible with contextual occupancy conditions. Since we predict a sequence of feature occupancies, the overall loss is computed across all frames. During contextual occupancy conditions, the $n_c$ noised latents are replaced by the ground truth (as explained above), and thus, the loss for those frames is ignored using the condition mask $\mathcal{M}$. The loss function for training the diffusion model is defined as:

$$\mathcal{L}_{\text{diffusion}} = \mathbb{E}_{t, \epsilon \in \mathcal{N}(0,1), i}\left[(1 - \mathcal{M}_t) \odot \left\| \epsilon_\theta\left(\hat{z}_i^t\right) - \epsilon \right\|^2\right] \quad (3)$$

where $\hat{z}_i^t$ is the $t$-th frame at diffusion timestamp $i$, and $\epsilon_\theta$ is the denoising network, specifically our DOME model.

## 3.3 TRAJECTORY AS CONDITIONING

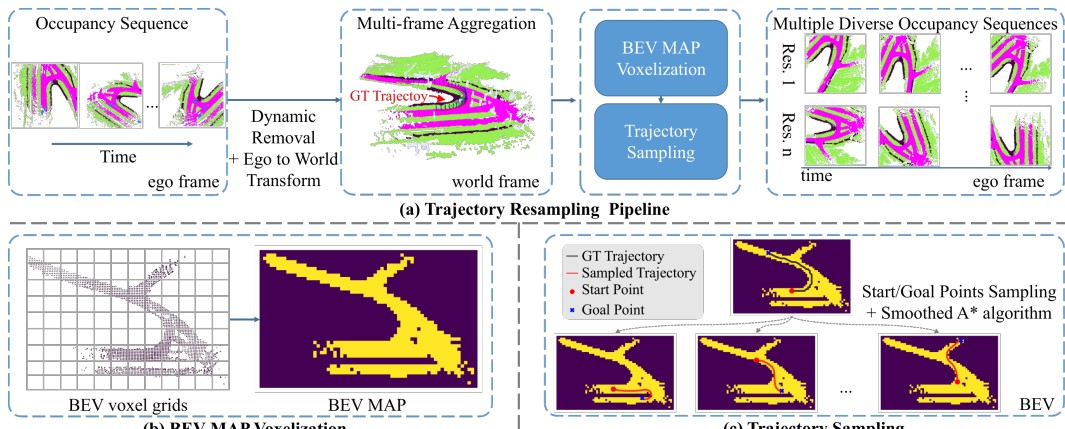

(a) Trajectory Resampling Pipeline

(b) BEV MAP Voxelization

(c) Trajectory Sampling

Figure 3: **(a) Trajectory Resampling Pipeline**: This process resamples multiple diverse and feasible occupancy sequences from a single ground-truth occupancy sequence. **(b) BEV Map Voxelization**: The road point clouds are voxelized into voxel grids to construct a BEV map representing the drivable area. **(c) Trajectory Sampling**: The smoothed A* algorithm is applied to generate multiple feasible trajectories on the BEV map.

**Trajectory Condition Injection**: Action conditioning is essential for world models, as the world observation $o^t$ should change coherently and reasonably based on the agent's last action $a^t$. We inject trajectory information into our model for conditional generation. Specifically, given the ego car's pose, we first calculate the relative translation $\Delta\mathbf{t}_t$ and relative rotation $\Delta\mathbf{R}_t$. From $\Delta\mathbf{t}_t$, we extract $[x, y] \in \mathbb{R}^{n_f \times 2}$, and from $\Delta\mathbf{R}_t$, we obtain the yaw angle $\theta_{\text{yaw}} \in \mathbb{R}^{n_f \times 1}$, representing the ego vehicle's heading. We then apply positional encoding (Mildenhall et al., 2020) to $[x, y, \theta_{\text{yaw}}]$, project the encoded values to the hidden size using a linear layer, and combine them with the time embedding. These combined values are then passed to the adaptive layer normalization (adaLN) block.

**Trajectory Resampling**: This issue stems from the imbalance and limited diversity in the training dataset. For example, in the nuScenes dataset (Caesar et al., 2019), the training set consists of 700 scenes, but the majority involve the vehicle moving straight (approximately 87%, see Fig. 4 (c)), highlighting the imbalance problem. Furthermore, in each scene, the vehicle only passes through once, resulting in a lack of diverse 3D occupancy samples under varying trajectory conditions within the same scene. This leads the model to overfit to the scenes, learning only the ground truth feature observations based on the contextual observation. The original trajectory distribution is shown in Fig. 4 (a).

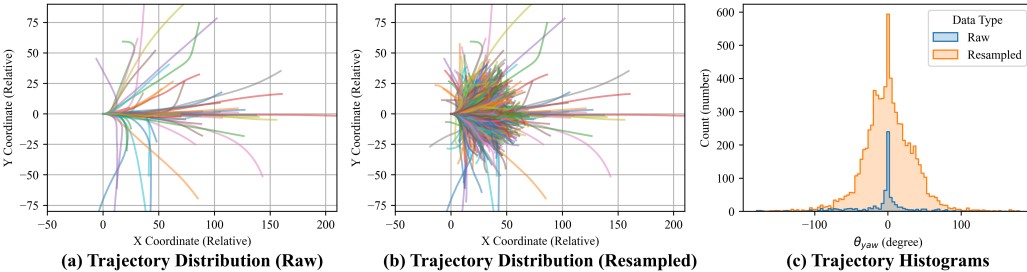

(a) Trajectory Distribution (Raw)

(b) Trajectory Distribution (Resampled)

(c) Trajectory Histograms

Figure 4: Trajectory distribution and histograms comparing the scenarios with and without trajectory resampling. For Figure (a) and (c), we use uniformly sampled trajectories from the dataset for better illustration and visualization.

To address this issue, we propose a trajectory resampling method, illustrated in Fig. 3 (a), with the corresponding pseudo-code provided in the Appendix. Our objective is to diversify the actions of the ego vehicle and the resulting sampled occupancy for each scene. The procedure consists of the following steps: (1) **Multi-frame Point Cloud Aggregation**: We start by converting the occupancy sequence in the ego frame into 3D point clouds, which are then transformed into the world frame

using the ego pose. Potential dynamic objects (e.g., cars, pedestrians) are filtered out by selecting based on the point cloud's semantic labels. (2) **Obtaining Drivable Area**: To generate diverse observations, we create various feasible trajectories based on the drivable area of the scene. After aggregating all point clouds into the world frame, we filter for road classes and voxelize the road point clouds from a top-down view to produce a Bird's Eye View (BEV) map (see Fig. 3 (b)). (3) **Generating Diverse and Feasible Trajectories**: Using the BEV map, we randomly sample two points representing the start and goal positions. We apply a smoothed A* algorithm (Hart et al., 1968) to generate a trajectory connecting these points, simulating the ego vehicle's driving trajectory. The resulting trajectory is converted into an $\mathbb{R}^{4\times4}$ pose, with the $z$ coordinate set to 0. (4) **Extracting Resampled Occupancy**: Using the trajectory pose, we apply an occupancy ground truth extraction method similar to that of Tian et al. (2023) to resample occupancy from the point cloud.

Our resampled trajectory distribution is illustrated in Fig. 4 (b). Compared to Fig. 4 (a), it fills the gaps in the trajectory distribution, demonstrating that our method enhances diversity and mitigates imbalance. This improvement is further supported by the driving direction histograms shown in Fig. 4 (c).

In conclusion, our trajectory resampling method is both simple and effective. To the best of our knowledge, we are the first to explore occupancy data augmentation for the task of world model prediction. This method is highly generalizable and can be applied to all types of occupancy data, including machine-annotated, LiDAR-collected, or self-supervised data. It requires only pose and occupancy data, without the need for LiDAR data or 3D bounding boxes.

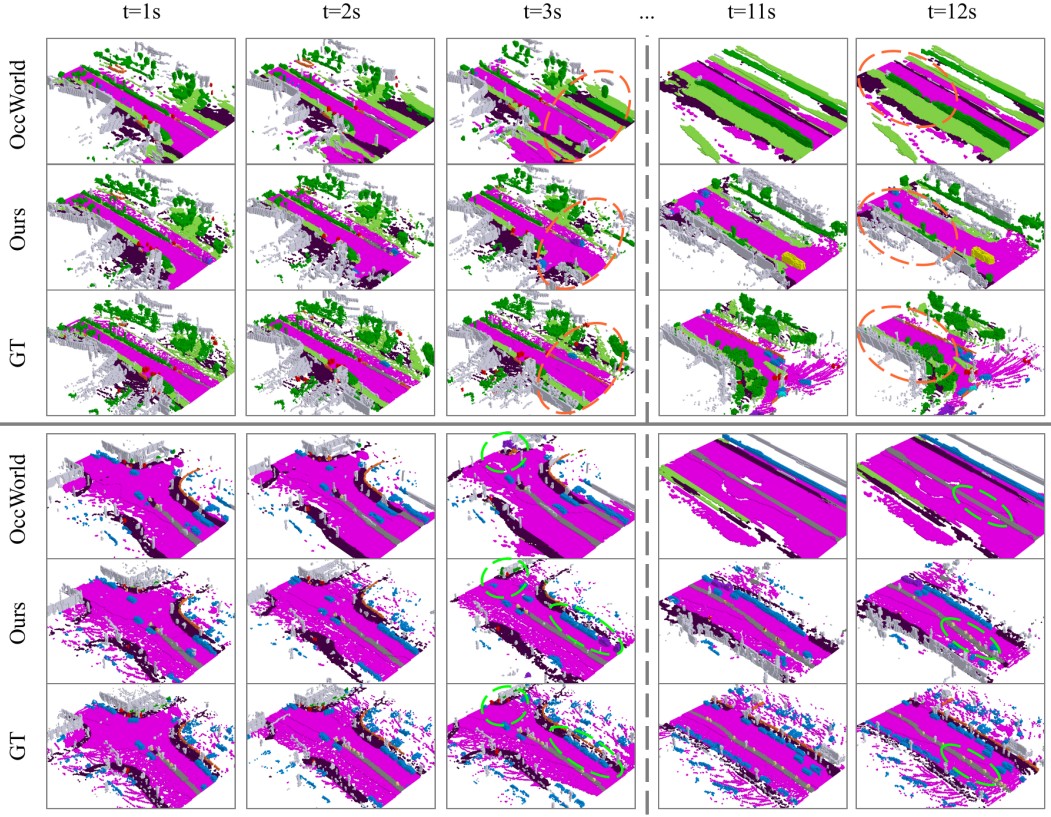

Figure 5: **Qualitative result of 4D occupancy forecasting.**

### 3.4 APPLICATIONS OF WORLD MODELS

**4D Occupancy Forecasting**: During inference, we begin with random noise corresponding to the buffer size of frames (the number of frames to be predicted) and encode $n_c$ contextual occupancy frames via Occ-VAE to obtain contextual latents. We replace the $n_c$ frames in the random noise with these contextual latents and then pass the input to our spatial-temporal DiT (see the bottom of Fig. 2). Throughout the denoising loop, the contextual latents remain unchanged as they are replaced in each iteration. After obtaining the denoised latent, we pass it to the Occ-VAE's decoder to generate the

final occupancy prediction. The hyperparameter $n_c$ can be adjusted based on different requirements. We set $n_c = 4$ for precise occupancy forecasting, as longer historical frames provide more scene and motion information. When greater controllability is needed, as dictated by the trajectory signal, we set $n_c = 1$ to reduce the influence of occupancy motion information while maintaining a controllable starting observation.

**Rollout for Long Duration Generation**: Due to limitations in computational resources and memory constraints, our model processes only $n_f$ frames of occupancy data for both training and inference. To generate longer occupancy predictions, we implement a rollout strategy similar to autoregressive approaches. Specifically, after generating the first $n_f$ frames, we reuse the last predicted frame as the contextual frame for predicting the next $n_f$ frames. An offset slices the corresponding trajectory to align with the contextual frame. This strategy can be applied iteratively to achieve long-term occupancy predictions.

# 4 EXPERIMENTS

## 4.1 EXPERIMENTAL SETUP

**Datsets**: We conduct our experiments on the widely used nuScenes dataset (Caesar et al., 2019), utilizing occupancy annotations from Occ3D (Tian et al., 2023). Following the setup of Zheng et al. (2023), we use the default training and validation settings, which include 700 and 150 occupancy sequences, respectively. Each occupancy sequence contains approximately 40 frames, sampled at a rate of 2 Hz. For each occupancy frame, the sample resolution is $[0.4, 0.4, 0.4]$ meters, covering a perception range of $[-40m, -40m, -1m, 40m, 40m, 5.4m]$, resulting in occupancy grids of size $[200, 200, 16]$. Each grid cell is assigned one of 17 semantic class labels based on LiDAR semantics.

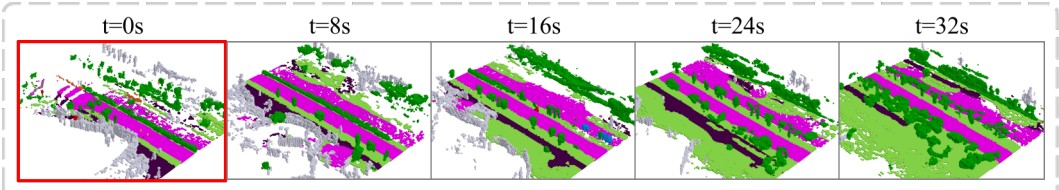

Figure 6: **Demonstration of long-duration generation capability.** Red borders indicate the condition frame.

**Evaluation Metric**: We use IoU and mIoU metrics for both Occupancy Reconstruction and 4D Occupancy Prediction. Higher IoU and mIoU values indicate reduced information loss during compression, reflecting better reconstruction performance and demonstrating a more accurate understanding of the surrounding environment for future predictions.

## 4.2 OCCUPANCY RECONSTRUCTION

Precisely reconstructing the occupancy while compressing it as much as possible is crucial for downstream tasks such as prediction and generation. Here, we compare Occ-VAE with existing methods that utilize an occupancy tokenizer and evaluate their reconstruction accuracy. The quantitative results of occupancy reconstruction are presented in Tab. 1. We achieve SOTA reconstruction performance for both IoU and mIoU metrics, with 83.1% for mIoU and 77.3% for IoU. Additionally, we have a relatively high compression rate of 64 times, being able to compress the occupancy data four times smaller than Zheng et al. (2023) and (Wei et al., 2024). Notably, we employ the same spatial compression rate (64 times) as described in Wang et al. (2024), but we differ in our approach by not applying the additional 8-times compression in the temporal dimension as they do. Instead, we strike a balance between compression and reconstruction performance. Moreover, excessive spatial downsampling would make contextual conditioning less inconvenient.

## 4.3 4D OCCUPANCY PREDICTION

We compare our method with existing 4D occupancy prediction approaches under various settings (Wei et al., 2024; Zheng et al., 2023). These settings include using ground-truth 3D occupancy data (-O) as input and using predicted results from off-the-shelf 3D occupancy predictors (-F). Following

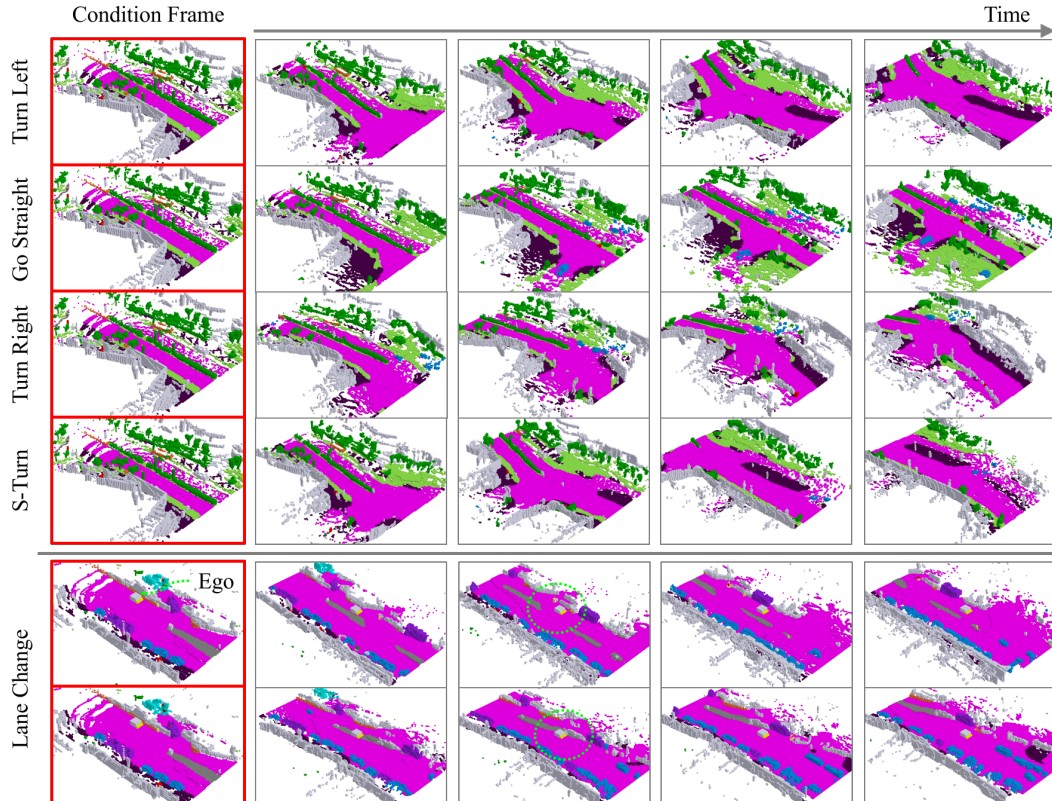

Figure 7: **Demonstration of different trajectory controls given the same contextual frame.** Red borders indicate the condition frame.

the experimental setup of Wei et al. (2024), we employ FB-OCC (Li et al., 2023b) as the occupancy extractor, utilizing predictions from camera input.

The qualitative results are shown in Fig. 5. The quantitative results shown in Tab. 2 indicate that our DOME-O achieves SOTA performance, with 27.10% for mIoU and 36.36% for IoU. We observe significant improvements over SOTA methods in both short-term (1s) and long-term (3s) predictions, demonstrating that our model effectively captures the fundamental evolution of the scene over time. The DOME-F can be considered an end-to-end vision-based 4D occupancy forecasting method, as it uses only surrounding camera captures as input. Despite the challenging nature of the task, our method achieves competitive performance, further demonstrating that DOME has strong generalizability

We also demonstrate our model's ability for long-duration generation, as shown in Fig. 6, and its capacity to be manipulated by trajectory conditions given the same starting frame, as illustrated in Fig. 7. Additionally, we compare our method's generation capability to existing occupancy world models in Tab. 4, where our approach shows the ability to generate the longest duration, achieving ten times the length of OccWorld and twice that of OccSora.

## 4.4 ABLATION STUDY

**Different Trajectory Condition**: We tested different settings of the trajectory condition, and the results are shown in Tab. 3. *Traj.* indicates whether or not to use the pose condition for prediction, *Res.* indicates whether or not to use our trajectory resampling enhancement, and *Yaw* indicates whether or not to add yaw angle embedding. Even without using any pose condition, we found that our model outperforms OccWorld (Zheng et al., 2023). Trajectory information significantly improves prediction by providing the model with a clear direction of scenario change, instead of requiring it to infer from multiple possibilities. The yaw angle embedding offers a slight improvement in IoU.

**Number of Contextual Frames**: We found that providing more contextual frames during the prediction process leads to better predictions (see Tab. 5), as additional frames give the model more explicit information about motion and changes in other vehicles and the scene. However, we also observed that increasing the number of frames is less efficient than using trajectory information, as

Table 1: **The quantitative analysis of occupancy reconstruction.**

| Method | Compression Ratio ↑ | mIoU ↑ | IoU ↑ | Others | barrier | bicycle | bus | car | const. veh. | motorcycle | pedestrian | traffic cone | trailer | truck | drive. suf. | other flat | sidewalk | terrain | man made | vegetation |
|---|---|---|---|---|---|---|---|---|---|---|---|---|---|---|---|---|---|---|---|---|
| OccWorld | 16 | 65.7 | 62.2 | 45.0 | 72.2 | 69.6 | 68.2 | 69.4 | 44.4 | 70.7 | 74.8 | 67.6 | 54.1 | 65.4 | 82.7 | 78.4 | 69.7 | 66.4 | 52.8 | 43.7 |
| OccSora | **512** | 27.4 | 37.0 | 11.7 | 22.6 | 0.0 | 34.6 | 29.0 | 16.6 | 8.7 | 11.5 | 3.5 | 20.1 | 29.0 | 61.3 | 38.7 | 36.5 | 31.1 | 12.0 | 18.4 |
| OccLLaMA | 16 | 75.2 | 63.8 | **65.0** | 87.4 | 93.5 | 77.3 | 75.1 | 60.8 | 90.7 | 88.6 | 91.6 | 67.3 | 73.3 | 81.1 | 88.9 | 74.7 | 71.9 | 48.8 | 42.4 |
| DOME (ours) | 64 | **83.1** | **77.3** | 36.6 | 90.9 | 95.9 | 85.8 | 92.0 | 69.1 | 95.3 | 96.8 | 92.5 | 77.5 | 86.8 | 93.6 | 94.2 | 89.0 | 85.5 | 72.2 | 58.7 |

Table 2: **4D occupancy forecasting performance.** Avg. denotes the average performance across 1s, 2s, and 3s. We use bold numbers to denote the best results. The suffix signifies different settings, with *-O* indicating that the input is occupancy. Other configurations first acquire occupancy through a 3D occupancy predictor before being input into the world model.

| Method | Input | mIoU (%) ↑ | | | | | IoU (%) ↑ | | | | |
|---|---|---|---|---|---|---|---|---|---|---|---|
| | | Recon. | 1s | 2s | 3s | Avg. | Recon. | 1s | 2s | 3s | Avg. |
| Copy&Paste | 3D-Occ | 66.38 | 14.91 | 10.54 | 8.52 | 11.33 | 62.29 | 24.47 | 19.77 | 17.31 | 20.52 |
| OccWorld-D | Camera | 18.63 | 11.55 | 8.10 | 6.22 | 8.62 | 22.88 | 18.90 | 16.26 | 14.43 | 16.53 |
| OccWorld-T | Camera | 7.21 | 4.68 | 3.36 | 2.63 | 3.56 | 10.66 | 9.32 | 8.23 | 7.47 | 8.34 |
| OccWorld-S | Camera | 0.27 | 0.28 | 0.26 | 0.24 | 0.26 | 4.32 | 5.05 | 5.01 | 4.95 | 5.00 |
| OccWorld-F | Camera | 20.09 | 8.03 | 6.91 | 3.54 | 6.16 | 35.61 | 23.62 | 18.13 | 15.22 | 18.99 |
| OccWorld-O | 3D-Occ | 66.38 | 25.78 | 15.14 | 10.51 | 17.14 | 62.29 | 34.63 | 25.07 | 20.18 | 26.63 |
| OccLLaMA-F | Camera | 37.38 | 10.34 | 8.66 | 6.98 | 8.66 | 38.92 | 25.81 | 23.19 | 19.97 | 22.99 |
| OccLLaMA-O | 3D-Occ | 75.20 | 25.05 | 19.49 | 15.26 | 19.93 | 63.76 | 34.56 | 28.53 | 24.41 | 29.17 |
| **DOME-F** (ours) | Camera | 75.00 | 24.12 | 17.41 | 13.24 | 18.25 | 74.31 | 35.18 | 27.90 | 23.435 | 28.84 |
| **DOME-O** (ours) | 3D-Occ | **83.08** | **35.11** | **25.89** | **20.29** | **27.10** | **77.25** | **43.99** | **35.36** | **29.74** | **36.36** |

the model must navigate ambiguous frame histories to predict future movements. This ambiguity is unnecessary for a world model that predicts scenes based on agent-determined movements.

Table 3: Ablation on key components. *Traj.* for trajectory, *Res.* for resampling augmentation.

| Spatio-Temp | Traj. Control | | | mIoU (%) ↑ | IoU (%) ↑ |
|---|---|---|---|---|---|
| | Traj. | Res. | Yaw | | |
| ✗ | ✗ | ✗ | ✗ | 13.08 | 23.10 |
| ✓ | ✗ | ✗ | ✗ | 18.60 | 28.09 |
| ✓ | ✓ | ✗ | ✗ | 24.24 | 34.28 |
| ✓ | ✓ | ✓ | ✗ | 27.00 | **36.39** |
| ✓ | ✓ | ✓ | ✓ | **27.10** | 36.36 |

Table 4: Comparison of generation durations across different methods.

| Method | Frame Rate | Frames ↑ | Duration (s) ↑ |
|---|---|---|---|
| OccWorld | 2Hz | 6 | 3 |
| OccLLaMA | 2Hz | 6 | 3 |
| OccSora | 2Hz | 32 | 16 |
| DOME (ours) | 2Hz | **64** | **32** |

Table 5: Ablation on different numbers of contextual frames and usage of trajectory.

| Cont. Frames | Traj. | mIoU (%) ↑ | IoU (%) ↑ | Cont. Frames | Traj. | mIoU (%) ↑ | IoU (%) ↑ |
|---|---|---|---|---|---|---|---|
| 1 | ✗ | 12.59 | 22.74 | 1 | ✓ | 22.24 | 32.71 |
| 2 | ✗ | 20.01 | 29.19 | 2 | ✓ | 25.41 | 35.00 |
| 3 | ✗ | 20.70 | 29.87 | 3 | ✓ | 26.61 | 36.14 |
| 4 | ✗ | 20.07 | 28.95 | 4 | ✓ | **27.10** | **36.36** |

## 5 CONCLUSION

In this paper, we propose DOME, a diffusion-based world model that forecasts future occupancy frames conditioned on historical data. It integrates Occ-VAE with a trajectory encoder and resampling technique to enhance controllability. To the best of our knowledge, we are the first to propose occupancy data augmentation for world model prediction. DOME demonstrates high-fidelity generation, effectively predicting future scene changes in occupancy space, and can generate long-duration occupancy sequences that are twice as extensive as those produced by previous methods. This approach holds promising applications for enhancing end-to-end planning in autonomous driving.

**Limitations and Future Work**. We found that training our model still requires significant computational resources. In the future, we will explore methods that are more lightweight and computationally efficient, or employ a fine-tuning paradigm to reduce resource requirements.

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

## A  APPENDIX

**Preliminaries of Diffusion Model**: We begin by revisiting the fundamental concepts of the diffusion model (Ho et al., 2020). A diffusion model consists of two processes: the noising process and the denoising process. During the noising process, Gaussian noise $\epsilon_i \sim \mathcal{N}(0, \mathbf{I})$ is gradually added to the real data sample $x_0$ to obtain the corrupted data $x_i$:

$$x_i = \sqrt{\bar{\alpha}_i} x_0 + \sqrt{1 - \bar{\alpha}_i} \epsilon_i,$$

where the granularity of the noise is controlled by the hyperparameter $\bar{\alpha}_i$. During the denoising process, the model learns to predict a denoised sample $x_{i-1}$:

$$p_\theta(x_{i-1} \mid x_i) = \mathcal{N}(\mu_\theta(x_i), \Sigma_\theta(x_i)).$$

The denoising process is optimized using the evidence lower bound (ELBO) (Kingma & Welling, 2013):

$$\mathcal{L}(\theta) = -p\left(x_0 \mid x_1\right) + \sum_i \mathcal{D}_{KL}\left(q^*\left(x_{i-1} \mid x_i, x_0\right) \| p_\theta\left(x_{i-1} \mid x_i\right)\right),$$

which can be simplified by calculating the mean squared error (MSE) between the predicted noise and the ground truth noise:

$$\mathcal{L}_{\text{simple}}(\theta) = \left\|\epsilon_\theta\left(x_i\right) - \epsilon_i\right\|_2^2.$$

**Spatial-temporal Forward Details**: When processing each spatial block $l_\phi$, the model treats the latent as a batch of separate patched images by integrating the temporal layers with the batch layers. When processing temporal blocks, the latent's spatial dimension is combined with the batch dimension.

This process can be written in `einops` (Rogozhnikov, 2022) notation as:

$$z' \leftarrow \text{rearrange}\left(z, (b, n_f, t, c) \rightarrow (b \times n_f, t, c)\right)$$
$$z' \leftarrow l_\theta^i\left(z', \mathbf{c}\right)$$
$$z' \leftarrow \text{rearrange}\left(z', (b \times n_f, t, c) \rightarrow (b \times t, n_f, c)\right)$$
$$z' \leftarrow l_\phi^i\left(z', \mathbf{c}\right)$$
$$z' \leftarrow \text{rearrange}\left(z', (b \times t, n_f, c) \rightarrow (b, n_f, t, c)\right)$$

where $b$ is the batch size dimension and $\mathbf{c}$ is the condition injected into DiT.

**Spatial and Temporal Positional Embedding**: After patchification, to enhance the model's understanding of spatial order, a ViT-style spatial positional embedding is applied to the tokens. The embedding weights are initialized using 2D sine and cosine functions and are fixed during training. This embedding is added to the spatial tokens across all temporal dimensions.

$$z_i' \leftarrow PE_{spatial} + z_i, \quad \forall i \in \{0, 1, \ldots, n_f\}$$

Where $PE_{spatial} \in \mathbb{R}^{t \times c}$ and $z_i \in \mathbb{R}^{t \times c}$. Similarly, we add positional embeddings to the temporal dimension to enhance the model's understanding of temporal correlations. We implement this using 1D sine and cosine functions, which are added across all spatial dimensions.

$$z_j' \leftarrow PE_{temperal} + z_j, \quad \forall j \in \{0, 1, \ldots, t\}$$

Where $PE_{temperal} \in \mathbb{R}^{n_f \times c}$ and $z_j \in \mathbb{R}^{n_f \times c}$.

**Trajectory Positional Encoding**: The function $\gamma$ is the positional encoding function, following the standard method of encoding positions using sine and cosine functions (Mildenhall et al., 2020):

$$\gamma(p) = \left(\sin\left(2^0 \pi p\right), \cos\left(2^0 \pi p\right), \cdots, \sin\left(2^{L-1} \pi p\right), \cos\left(2^{L-1} \pi p\right)\right)$$

**Trajectory Resampling Pseudo Code**:

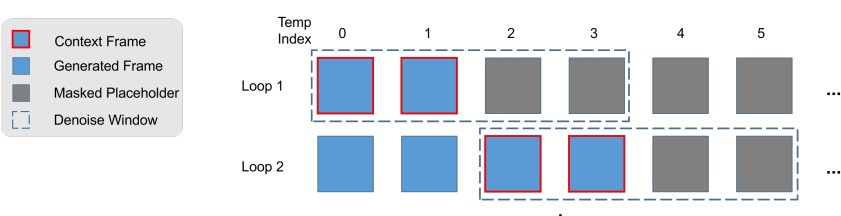

Figure 8: **Rollout Generation Demonstration**.

---

**Algorithm 1** Trajectory Resampling Method

---

1: **Input:** Occupancy sequence in ego frame
2: **Output:** Diversified actions and sampled occupancy
3: **procedure** TRAJECTORYRESAMPLING(occupancySequence, egoPose, numSamples)
4:     $pointClouds \leftarrow$ AggregatePointClouds(occupancySequence)
5:     $worldPointClouds \leftarrow$ TransformToWorldFrame(pointClouds, egoPose)
6:     $filteredClouds \leftarrow$ FilterDynamicObjects(worldPointClouds)
7:     $drivableArea \leftarrow$ GenerateDrivableArea(filteredClouds)
8:     $BEVMap \leftarrow$ VoxelizeRoadClasses(drivableArea)
9:     $resampledOccupancy \leftarrow$ InitializeEmptyList()
10:     **for** sample in 1 to $numSamples$ **do**
11:         $(start, goal) \leftarrow$ RandomSamplePoints(BEVMap)
12:         $trajectory \leftarrow$ SmoothedAStar(start, goal)
13:         $pose \leftarrow$ ConvertToPose(trajectory)
14:         $occupancy \leftarrow$ ExtractOccupancy(pose)
15:         $resampledOccupancy$.Append($occupancy$)
16:     **end for**
17:     **return** $resampledOccupancy$
18: **end procedure**

---

**Rollout for Long Duration Generation**: Our rollout strategy is illustrated in Fig. 8. Each time, the world model predicts the content of the masked region within a denoising window. Once the denoising loop concludes, we replace the masked section with the content generated during denoising. In the subsequent denoising loop, the content from the end of the previous prediction is used as context. This process continues iteratively until all placeholders have been predicted.

**Implementation Details**: We train Occ-VAE using AdamW with a learning rate of $1 \times 10^{-3}$ and a cosine scheduler on input shapes $200 \times 200 \times 16$, with a batch size of 10 per GPU for 200 epochs on 8 RTX 4090 GPUs. For the second and third stages, we use the model with 14 spatial and 14 temporal layers, trained with AdamW and EMA, at a batch size of 8 per GPU for 2000 epochs on 32 RTX 4090 GPUs. We employ `xformers`, mixed precision, and gradient checkpointing to reduce memory usage. We use DDPM with 1000 diffusion steps for training and 20 steps for inference. We set $n_f = 11$, $n_c = 4$, $n_h = n_w = 25$, and the patch size $p = 1$.

**Visualization of 4D Occupancy forecasting samples.** We demonstrate our 4D occupancy forecasting samples here. The red-bordered box indicates the conditional frames, while the remaining frames are the predicted forecasts.

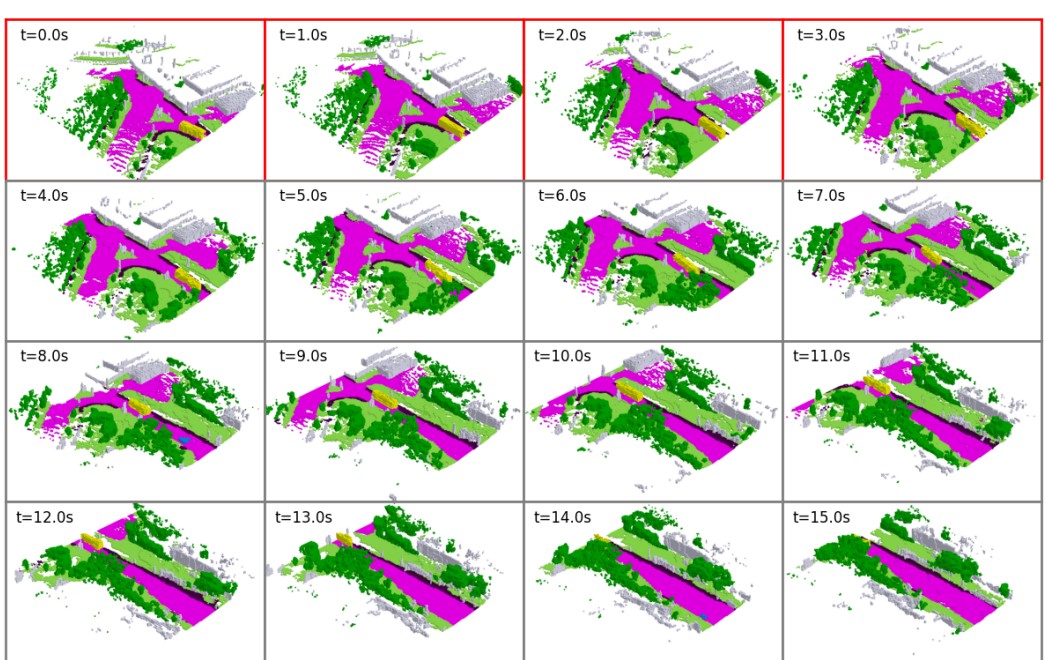

Figure 9: **4D Occupancy forecasting samples.** Red borders indicate the condition frame.

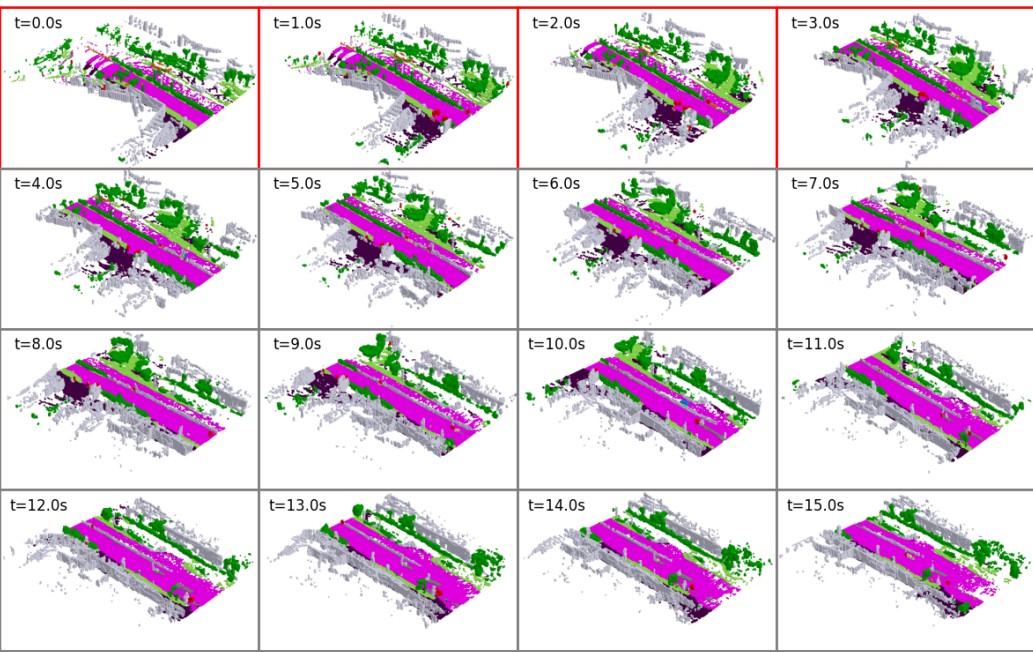

Figure 10: **4D Occupancy forecasting samples.** Red borders indicate the condition frame.

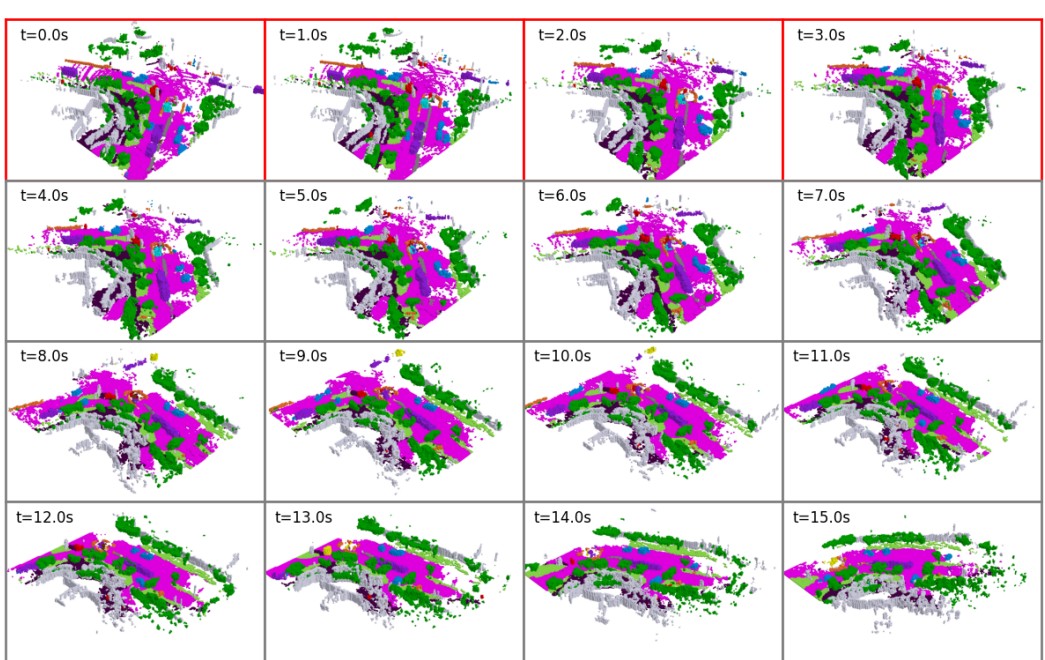

Figure 11: **4D Occupancy forecasting samples.** Red borders indicate the condition frame.

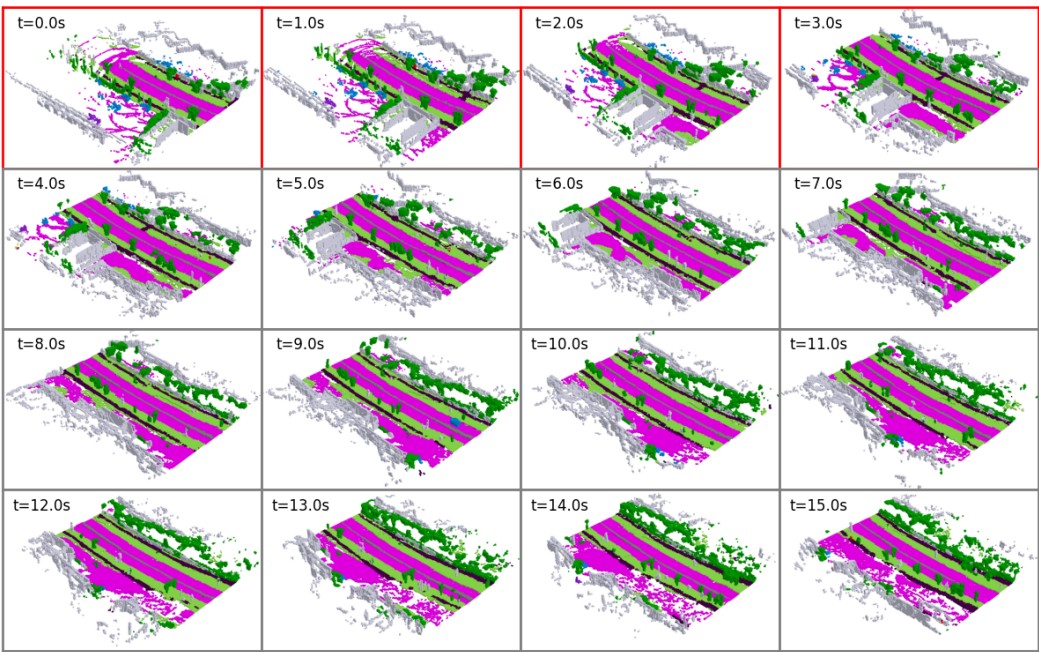

Figure 12: **4D Occupancy forecasting samples.** Red borders indicate the condition frame.

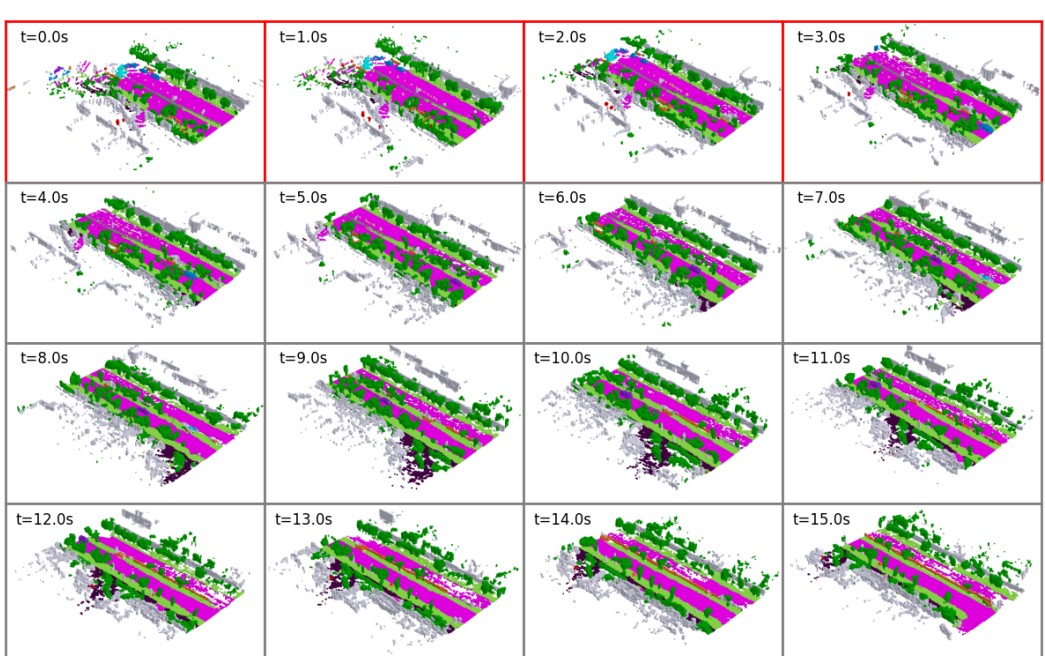

Figure 13: **4D Occupancy forecasting samples.** Red borders indicate the condition frame.

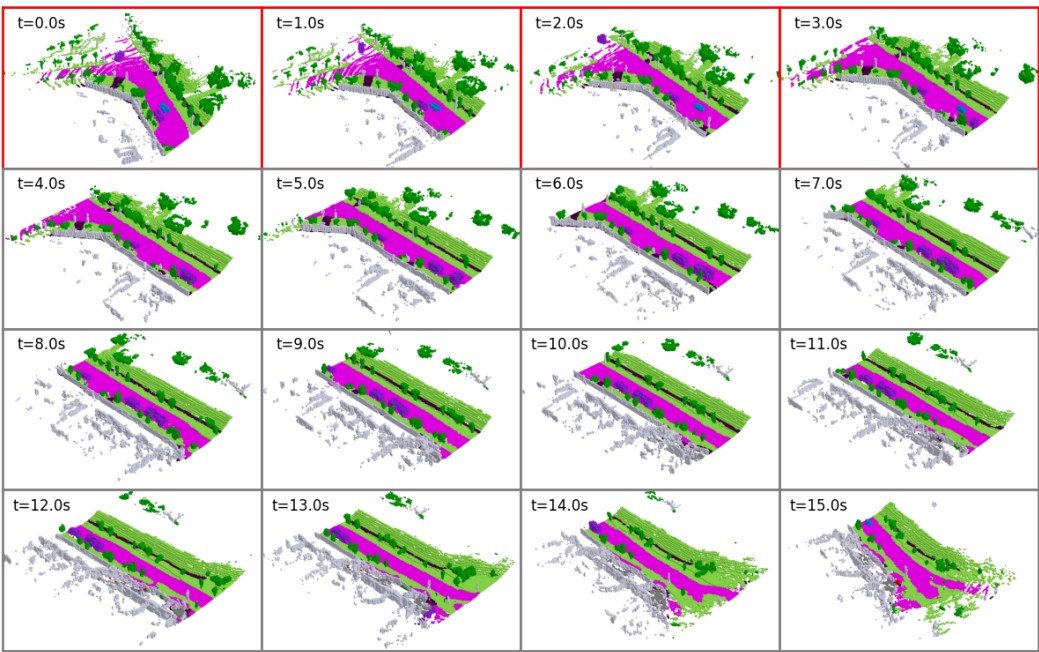

Figure 14: **4D Occupancy forecasting samples.** Red borders indicate the condition frame.

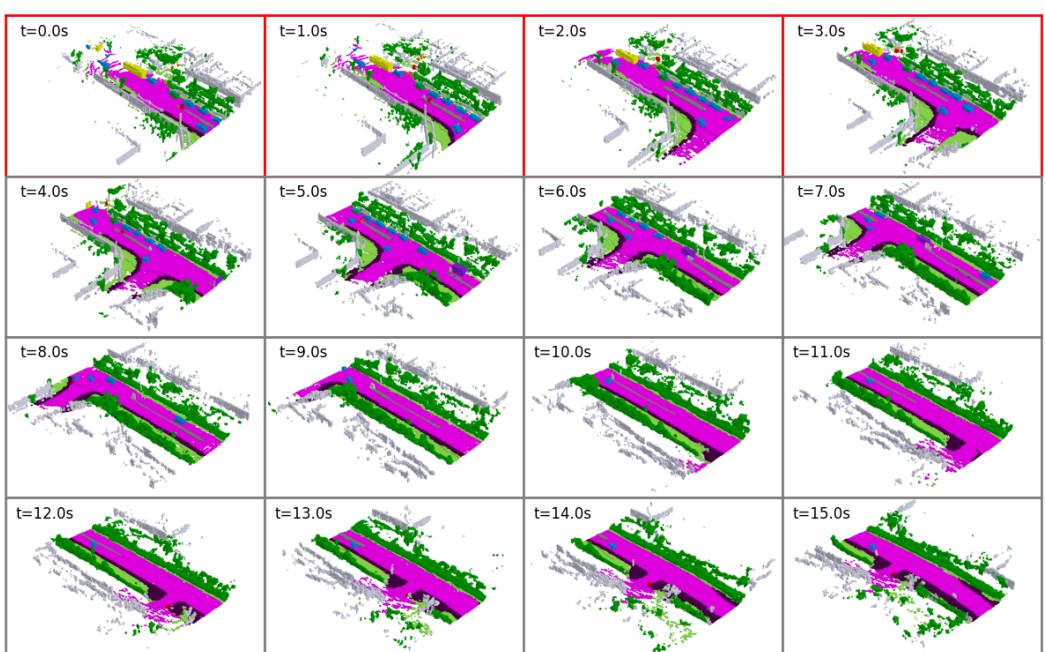

Figure 15: **4D Occupancy forecasting samples.** Red borders indicate the condition frame.

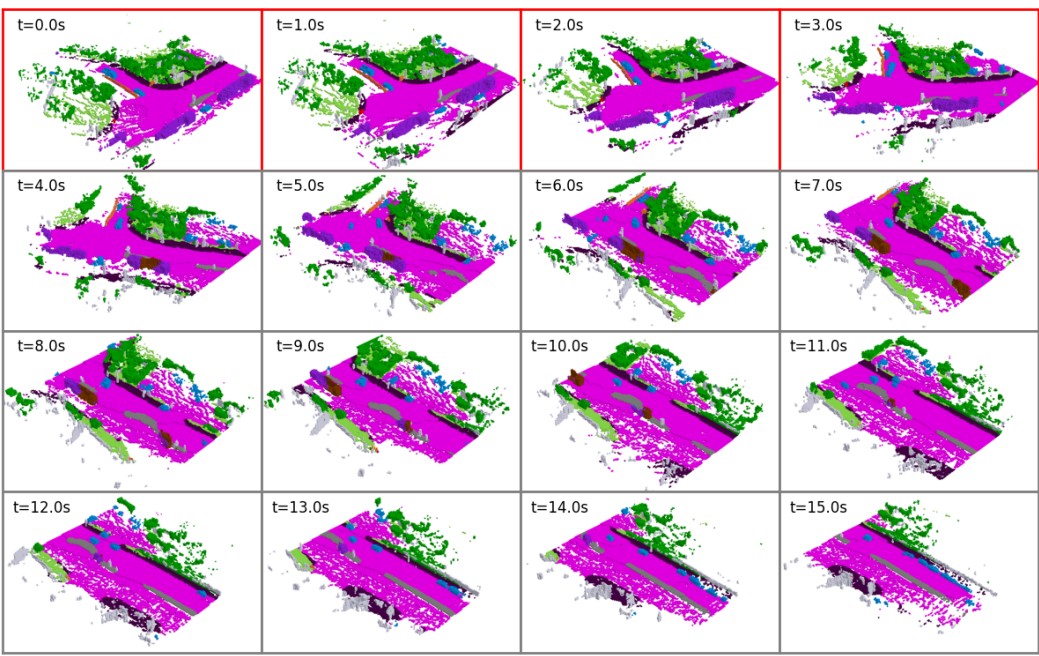

Figure 16: **4D Occupancy forecasting samples.** Red borders indicate the condition frame.

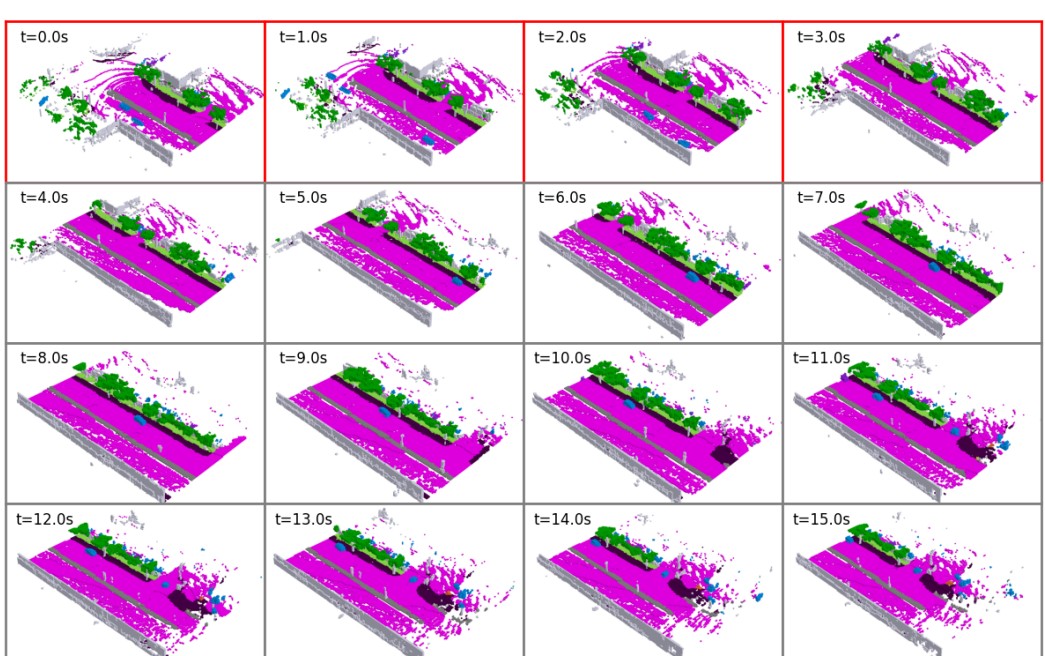

Figure 17: **4D Occupancy forecasting samples.** Red borders indicate the condition frame.

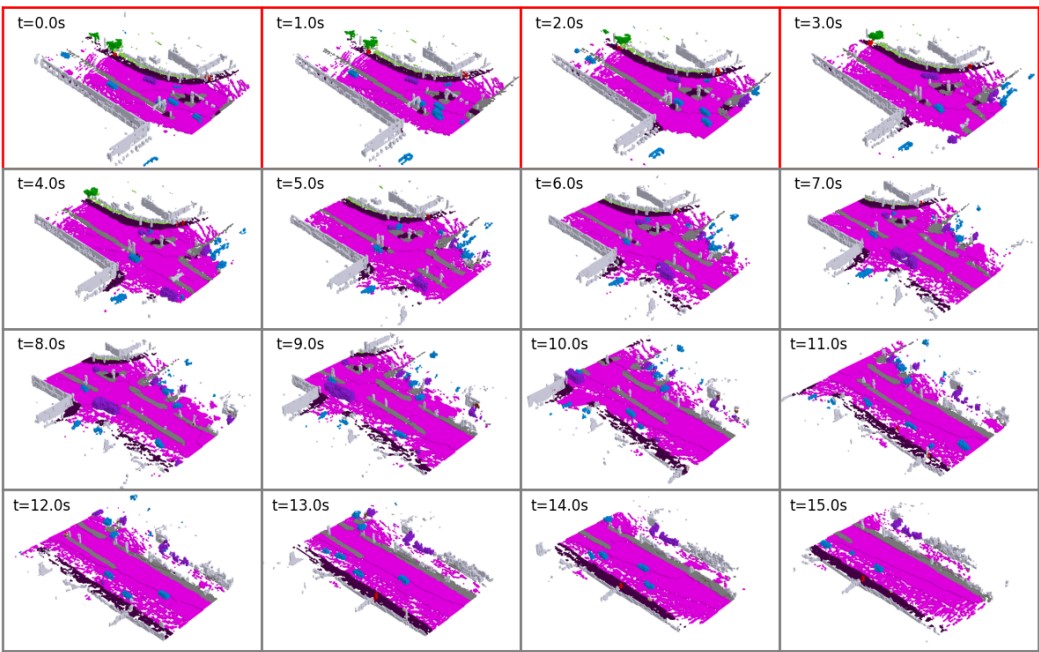

Figure 18: **4D Occupancy forecasting samples.** Red borders indicate the condition frame.

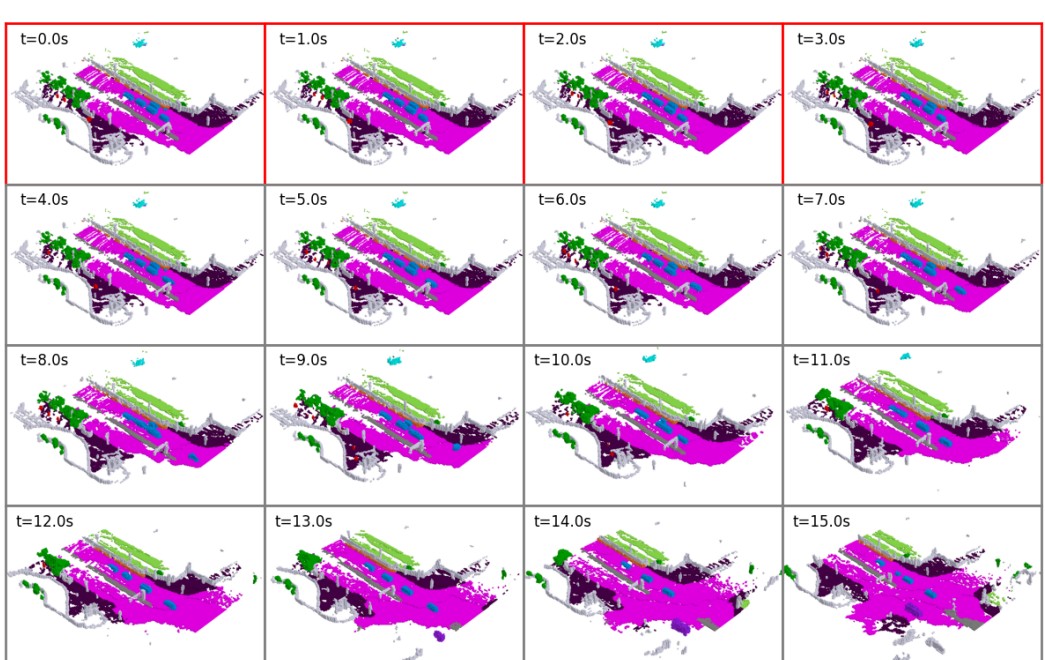

Figure 19: **4D Occupancy forecasting samples.** Red borders indicate the condition frame.

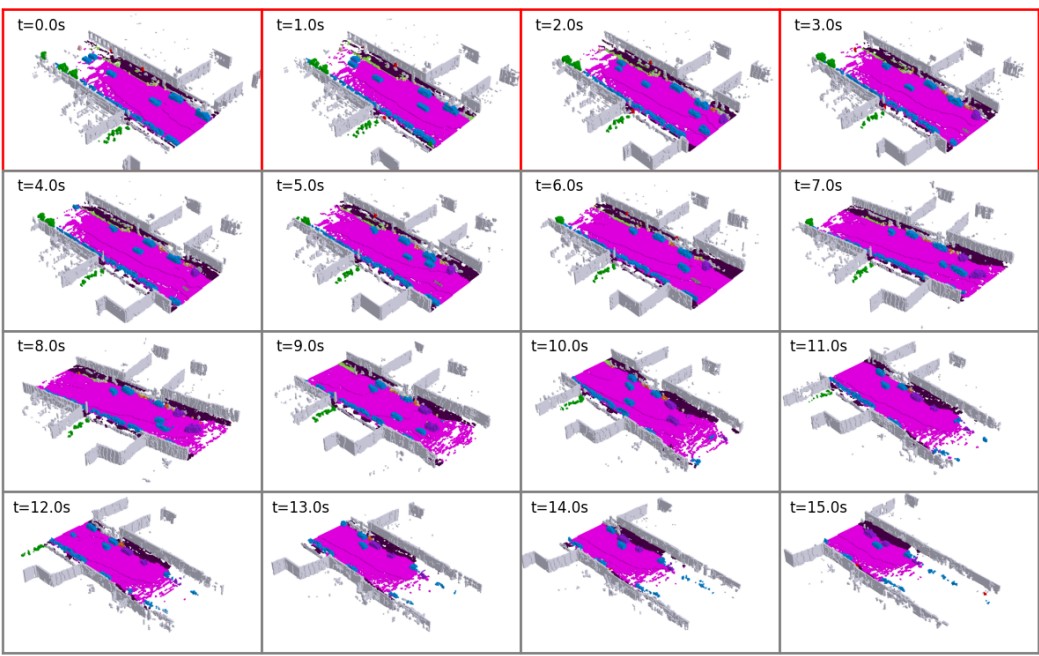

Figure 20: **4D Occupancy forecasting samples.** Red borders indicate the condition frame.

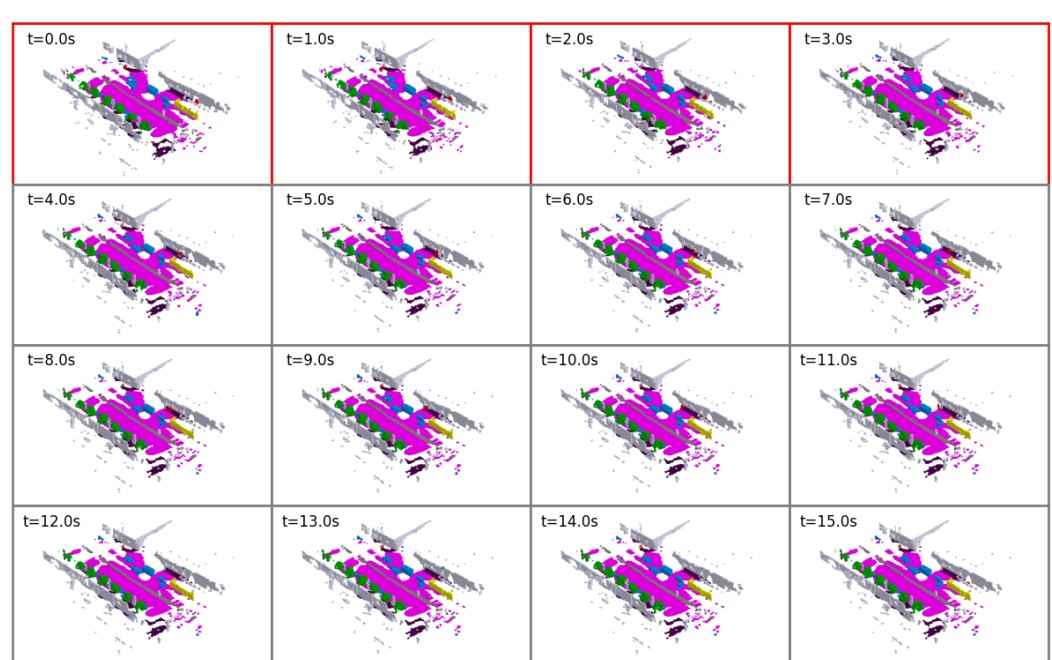

Figure 21: **4D Occupancy forecasting samples.** Red borders indicate the condition frame.

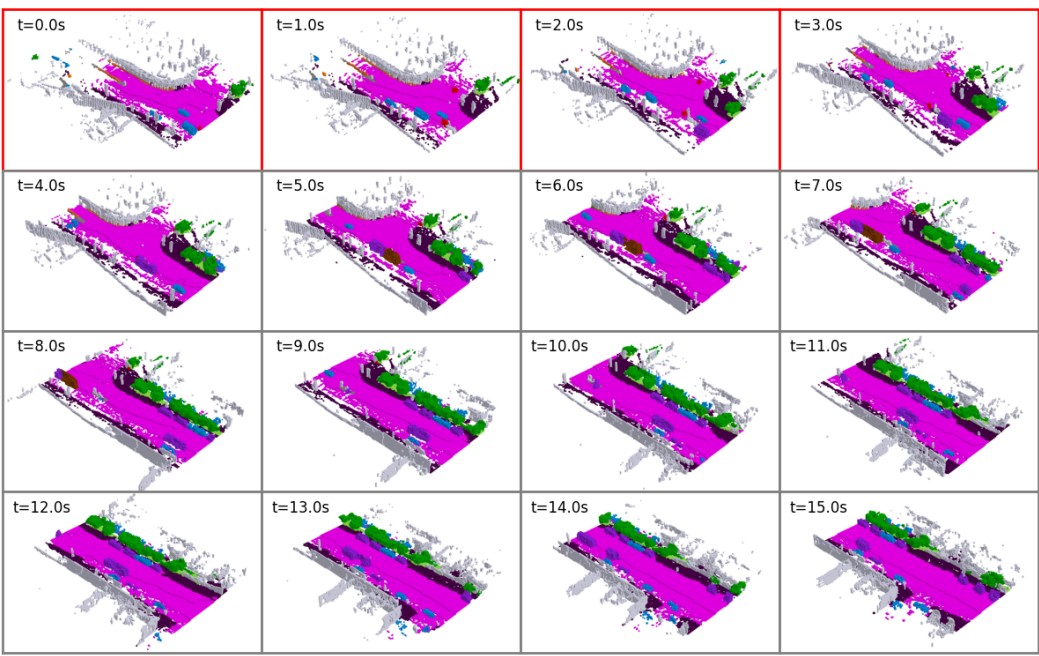

Figure 22: **4D Occupancy forecasting samples.** Red borders indicate the condition frame.

