# OpenReview forum: "DOME: Taming Diffusion Model into High-Fidelity Controllable Occupancy World Model"
_ICLR.cc/2025/Conference — Submitted to ICLR 2025_

### Official Review · Reviewer_YnqS · 2024-10-30

**Soundness:** 2
**Presentation:** 2
**Contribution:** 2
**Rating:** 5
**Confidence:** 4

**Summary:**

The paper introduces DOME, a novel diffusion-based world model for autonomous driving that predicts future occupancy frames from past observations. DOME features high-fidelity, long-duration generation, and fine-grained controllability through a spatial-temporal diffusion transformer and a trajectory resampling method. Experiments on the nuScenes dataset demonstrate DOME's state-of-the-art performance, surpassing baselines by significant margins in both occupancy reconstruction and forecasting. The model's ability to generate long-duration sequences and respond to trajectory conditions could enhance end-to-end planning in autonomous driving.

**Strengths:**

1. DOME utilizes a spatial-temporal diffusion transformer that efficiently captures both spatial and temporal information, enabling the generation of detailed, long-lasting occupancy predictions.
2. DOME enhances the model's ability to generate controlled predictions, which is an important property for world models.
3. Extensive experiments on the nuScenes dataset show that DOME surpasses existing baselines by a considerable margin in both qualitative and quantitative evaluations for occupancy reconstruction and forecasting.

**Weaknesses:**

1. DOME's sophisticated architecture, while effective, might lead to increased complexity in implementation and maintenance. This complexity could make it harder for other researchers to reproduce the results or integrate the model into their own systems without a deep understanding of the underlying mechanisms.
2. The control of directions is still not accurate. From the visualizations, the ego car just slightly turns left or right, given the high-level command. Also, the generated scenes (e.g., the buildings and trees) are different for the same input scene with different trajectory conditions. I think a good world model should be able to generate consistent results for different trajectory conditions.
3. The authors do not compare their methods with OccSora (visualizations), which can also achieve long-term and trajectory-conditioned generation.
4. Despite the trajectory resampling method aimed at enhancing diversity, the paper does not extensively discuss the potential for overfitting, especially given the model's complexity. More analysis on how the model performs on unseen data would strengthen the paper's claims.

**Questions:**

Can you provide a visualization comparison with OccSora about trajectory conditions?

---

> ### Author Response · Authors · 2024-11-21
>
> We thank the reviewer for highlighting DOME's strengths, including its spatial-temporal diffusion transformer, enhanced controllability, and strong performance on the nuScenes dataset, surpassing baselines in both qualitative and quantitative evaluations. We address any further comments below.
>
> ---
>
> > 1. Q: "DOME's sophisticated architecture, while effective, might lead to increased complexity in implementation and maintenance. This complexity could make it harder for other researchers to reproduce the results or integrate the model into their own systems without a deep understanding of the underlying mechanisms."
>
> **A:**  We respectfully disagree with this concern. Firstly, we believe our architecture is **not overly complex**. It comprises a VAE and a diffusion-based generation model, both are standard designs in generation models.
>
> Additionally, to further support **reproducibility and integration**, we will be **open-sourcing our code soon**, enabling other researchers to easily access, understand, and build upon our work.
>
> ---
>
> > 2. Q:"The control of directions is still not accurate. From the visualizations, the ego car just slightly turns left or right, given the high-level command."
>
> **A:**  We would like to emphasize that **precise trajectory control** is a particularly challenging task, especially for generation models. Previous works either **did not address control** or implemented **trajectory conditioning naively**, which performed poorly in **turning scenarios** (as shown in the below sample). We believe our method achieves more effective trajectory control compared to other methods.
>
> - [turning_case_comparison](https://anonymous.4open.science/r/iclr-rebuttal-7BC0/turning_case_comparison.mp4)
>
> We aim to further validate our trajectory accuracy by **demonstrating multi-frame occupancy registration**, where we register multiple frame occupancies into the world frame based on their poses. This method is often adopted in LiDAR odometry to verify pose accuracy.
>
> - [multi_frame_register](https://anonymous.4open.science/r/iclr-rebuttal-7BC0/multi_frame_register.mp4)
>
> ---
>
> > 3. Q:" the generated scenes (e.g., the buildings and trees) are different for the same input scene with different trajectory conditions. I think a good world model should be able to generate consistent results for different trajectory conditions."
>
> **A:**  We respectfully disagree with this concern. **Generation models are inherently stochastic**, often producing varied outputs even under the same conditions, let alone under different trajectory conditions, as demonstrated in our results.
>
> ---
>
> > 4. Q:"The authors do not compare their methods with OccSora (visualizations), which can also achieve long-term and trajectory-conditioned generation."
>
> **A:** We would like to point out that OccSora should be considered a **contemporary work** as it is also under submission to ICLR 2025 and has not yet been published in **peer-reviewed conference proceedings or journals** (see [ICLR’s guidelines](https://iclr.cc/Conferences/2025/ReviewerGuide#Reviewer)).  So we are not required to compare our method with it.
> Additionally, we found that **OccSora's checkpoint is not publicly available**, making a direct comparison infeasible even if we wanted to include it.
>
> ---
>
> > 5. Q:"Despite the trajectory resampling method aimed at enhancing diversity, the paper does not extensively discuss the potential for overfitting, especially given the model's complexity. More analysis on how the model performs on unseen data would strengthen the paper's claims."
>
> **A:**  We would like to clarify that **our results were tested on the test set**, following the settings of prior works. This experiment has sufficiently validated our method’s generalization.
> Nevertheless, as requested by the reviewer, **we have also tested our model on other datasets** (KITTI), and the results are presented below. These results demonstrate that our model performs well on unseen data, and we hope this addresses the reviewer’s concerns about overfitting.
>
> - [kitti_demo](https://anonymous.4open.science/r/iclr-rebuttal-7BC0/kitti.mp4)

---

> ### Author Response · Authors · 2024-11-26
>
> Since the discussion stage is nearing its end, we would appreciate your feedback and are happy to address any concerns you may have.

---

### Official Review · Reviewer_JrJ2 · 2024-11-02

**Soundness:** 2
**Presentation:** 2
**Contribution:** 2
**Rating:** 5
**Confidence:** 4

**Summary:**

This paper proposes a diffusion-based world model designed to predict future occupancy frames based on past occupancy observations. The authors introduce a spatial-temporal diffusion transformer, which leverages historical context to forecast future occupancy frames, effectively capturing spatiotemporal information for high-fidelity detail and enabling long-duration predictions. Additionally, the authors present a trajectory resampling method to address controllability challenges in prediction, significantly enhancing the model's ability to generate controlled predictions. Through experiments on the nuScenes dataset, the authors demonstrate that their approach outperforms existing baselines in both qualitative and quantitative evaluations, establishing new state-of-the-art performance on the nuScenes dataset. Specifically, their method surpasses the baseline by 10.5% in mIoU and 21.2% in IoU for occupancy reconstruction, and by 36.0% in mIoU and 24.6% in IoU for 4D occupancy prediction.

**Strengths:**

1. The authors propose a spatial-temporal interactive VQ encoder that facilitates spatiotemporal information exchange, contributing to higher-accuracy predictive generation.
2. They introduce a trajectory resampling method that reconstructs trajectories using the A* algorithm and designs possible scenarios based on a BEV map, improving the relationship between control and scene.
3. Experiments on the nuScenes dataset show that their approach achieves superior performance in both occupancy reconstruction mIoU and 4D occupancy prediction compared to existing baselines.

**Weaknesses:**

1. The spatial-temporal diffusion transformer lacks strong innovation; leveraging spatial-temporal information fusion and cross-attention is common in existing works, making this a less distinctive contribution.
2. The proposed trajectory resampling method, which reconstructs possible scenes based on a BEV map, may not adequately address dataset imbalance and limited diversity. The dataset used is an autonomous driving dataset with ego-vehicle movement, not a static city-scale occupancy dataset, which limits the effectiveness of trajectory resampling in addressing these issues.
3. The experiments for occupancy reconstruction and 4D occupancy prediction lack fairness in comparison. The reconstruction phase includes multi-frame inputs and spatio-temporal interaction, and the 4D occupancy prediction also leverages training parameters from the reconstruction stage. The authors should provide a fair comparison with other multi-frame input methods in the experimental section.

**Questions:**

1. In Figure 4(a), the authors highlight the limited trajectory diversity in the original dataset, while Figure 2 in OccLLaMA [1] demonstrates a more diverse range of trajectories, so whether the dataset is indeed as sparse as described in the paper.

2. The authors claim to achieve occupancy predictions under different trajectory controls;   however, the visualized results appear somewhat limited, as the vehicle's movement along the trajectory is mini  Could the authors present a 32-second controlled trajectory prediction and provide a clearer visualization of the relationship between trajectory control and the scene?

3. How does the author solve the problem of dynamic objects in the reconstruction of possible scenes by using BEV map in trajectory resamping? The nuScenes data set will move with the self-driving car, and the surrounding vehicles will also move. It seems difficult to make a reasonable dynamic scene by changing the track of a self-driving vehicle (i.e. maintaining consistency between multiple different vehicle resampling tracks and their respective Occupancy dynamic scenes).

[1] Wei, Julong, et al. "OccLLaMA: An Occupancy-Language-Action Generative World Model for Autonomous Driving." arXiv preprint arXiv:2409.03272 (2024).

---

> ### Author Response · Authors · 2024-11-21
>
> We thank the reviewer for recognizing our contributions, including the spatial-temporal interactive VQ encoder, the trajectory resampling method leveraging A* and BEV maps, and the strong experimental results on the nuScenes dataset. We address any further comments below.
>
> ---
>
> > 1. Q:"The spatial-temporal diffusion transformer lacks strong innovation; leveraging spatial-temporal information fusion and cross-attention is common in existing works, making this a less distinctive contribution."
>
> **A:** We respectfully disagree with this statement. We would like to emphasize that **we are the first to successfully apply this approach to the task of an occupancy world model**. Our work distinguishes itself from others in terms of representation. While previous spatial-temporal diffusion transformers primarily handle video data, we address 3D sequences, which are significantly more challenging. To learn this effectively, we design a more efficient latent compression method to alleviate computational costs.
>
> ---
>
> > 2. Q:“The proposed trajectory resampling method may not adequately address dataset imbalance and limited diversity. The dataset used is an autonomous driving dataset with ego-vehicle movement, not a static city-scale occupancy dataset, which limits the effectiveness of trajectory resampling in addressing these issues.”
>
>   **A:** We found that the **imbalance and limited diversity primarily arise from trajectory types** (with more trajectories going straight than turning), the diversity of scenes is sufficient in terms of our task. Furthermore, our experiments show that trajectory resampling **effectively improves controlled generation in our task**.
>
>   We would like to clarify that **static city-scale occupancy datasets are difficult to obtain in real** applications. Most real-world datasets (e.g., nuScenes, Waymo, KITTI) are collected by vehicles equipped with sensors. Hence, our method is more likely to be adopted in real autonomous driving scenarios.
>
>
> ---
>
> > 3. Q:"The comparison of experiments for occupancy reconstruction and 4D occupancy prediction lacks fairness, as it includes multi-frame inputs and spatio-temporal interactions."
>
> **A:**  We would like to clarify that our approach ensures **fairness in comparison**, as all methods use the **same number of input frames (4 frames)** during inference. Our method leverages spatio-temporal interaction **to fully utilize temporal information from multi-frame inputs** during the training of the VAE. This enables better temporal coherence given the same multi-frame inputs in inference.
>
> ---
> > 4. Q:"Why does Figure 4(a) show limited trajectory diversity compared to the more diverse range in OccLLaMA? Is the dataset truly as sparse as described in the paper?"
>
> **A:**  We want to clarify that in Figure 4(a), we **plotted only a uniformly sampled subset of trajectories just for reference**. As it would be less informative to represent true diversity. For  example, most trajectories are straight, and there will overlap in the figure. About the diversity of trajectory, we did a statistic that found that **approximately 87% of the training trajectories in our training set are going straight**.
>
> To avoid any misunderstanding, we have added a clarification about this sampling method in the figure caption in the revised paper.
>
> ---
>
> > 5. Q:"The authors claim to achieve occupancy predictions under different trajectory controls; however, the visualized results appear somewhat limited, as the vehicle's movement along the trajectory is mini Could the authors present a 32-second controlled trajectory prediction and provide a clearer visualization of the relationship between trajectory control and the scene?"
>
> **A:**  We would like to clarify that **we have already provided 9 visualized results in the main paper, 14 in the appendix, and 20 in the supplementary videos**. Please refer to these for further details. Additionally, we provide an extra demo of a 32-second prediction and its corresponding trajectory below.
>
> - [4D occupancy](https://anonymous.4open.science/r/iclr-rebuttal-7BC0/long_seq_w_pose.mp4)
> - [corresponding trajectory](https://anonymous.4open.science/r/iclr-rebuttal-7BC0/pose_w.png)
>
>
> ---
> > 6. Q:"How does the author solve the problem of dynamic objects in the reconstruction of possible scenes by using BEV map in trajectory resamping? "
>
> **A:**  Thank you for pointing out this concern. Currently, we **directly remove dynamic foreground objects** during trajectory resampling, as the primary motivation for adopting trajectory resampling is to address turning cases and enhance controllability. These factors are more closely related to **background static classes** (e.g., drivable areas and sidewalks). Our experiments demonstrate that using static classes alone for trajectory resampling **is sufficient to achieve good results**.

---

> ### Author Response · Authors · 2024-11-26
>
> Since the discussion stage is nearing its end, we would appreciate your feedback and are happy to address any concerns you may have.

---

> ### Comment · Reviewer_JrJ2 · 2024-11-26
> **Official Comment by Reviewer JrJ2**
>
> Thank you to the authors for their efforts in addressing my questions.
>
> Regarding A1:
>
> I suggest that the authors place greater emphasis on the unique aspects of their proposed method while providing a concise introduction to existing methods. This would better highlight the innovations of DOME. Additionally, while the authors state that "handling 3D sequences is much more challenging," it is worth noting that compared to RGB image/video sequences, the number of categories in 3D sequences is smaller. It might be important to clarify in the paper why this task is "much more challenging."
>
> Regarding A2&A6:
>
> I remain deeply concerned about the trajectory resampling method that removes foreground objects, as it may not adequately address data imbalance and limited diversity, particularly in the context of autonomous driving world models. A key function of such models is to enhance decision-making and planning by enabling systems to predict multiple possible futures and select the optimal path. If dynamic foreground objects—such as vehicles and pedestrians—are removed, the occupancy world model's relevance to planning and decision-making may be significantly diminished. The resampling approach may primarily expand static scenes, but excluding data on vehicles and pedestrians makes it challenging to effectively support the core tasks required of autonomous driving world models.
>
> Regarding A4:
>
> Thank you for the response. Drawing a uniformly sampled subset of trajectories can indeed lead to misunderstandings, particularly in how readers assess the true role and significance of evaluating data imbalance. I agree with the authors that improvements in this regard would be beneficial.
>
> General:
>
> I sincerely appreciate the authors' response. However, I remain very concerned about the trajectory resampling method that removes dynamic foreground objects in order to expand the dataset for autonomous driving occupancy world models. While this resampling approach better associates background and vehicle trajectories, it may lose the importance of the occupancy world model's ability to improve decision-making and planning. It could even result in a model misunderstanding the true occupancy by training on an incrementally expanded dataset.

---

> > ### Author Response · Authors · 2024-11-26
> >
> > Thank you for the insightful comments and thought-provoking concerns. We will address your questions below.
> >
> > > Q1: Challenges of 3D sequence generation compared to video generation.
> >
> > **A:** There are several challenges in 3D sequence generation compared to video generation:
> > (a). 3D occupancy often contains a lot of empty space and is more irregular and sparse, whereas video is regular and dense.
> >
> > (b). Compared to video sequences, 3D scenes have additional high-dimensional. Although the elements of a 3D scene may contain fewer categories, they involve more spatial elements.
> >
> > (c). 3D scenes often contain high-frequency details, such as small objects (e.g., traffic signs, trash bins), as well as discrete elements caused by the sparse regions of LiDAR data.
> >
> > (d). While video generation has been studied to some extent, 3D scene sequence generation is less explored due to the lack of large, transferable datasets and pre-trained models.
> >
> > Overall, we believe that 3D sequence generation is more challenging and warrants further exploration.
> >
> > ---
> >
> > > Q2:  Concerned about the trajectory resampling method that removes dynamic foreground objects.
> >
> >
> > **A:** We would like to clarify that we did not train our model solely with augmented data. Instead, we trained it using both the original data, which contains foreground objects, and the augmented data, with a 1:1 proportion of each. Our experiments show that this approach does not cause foreground objects to disappear significantly. **Overall, the trajectory resampling process does not degrade the quality of dynamic objects.**
> >
> > Secondly, the introduction of trajectory resampling is to alleviate the loss of control problem caused by **imbalance trajectory and the limited trajectories per scene**, rather than diverse 3D scenes. **Our experiment shows that our method achieves better control results.** To further clarify the misunderstanding, we have improved our writing on this in the revised paper.
> >
> > We also agree with the reviewer that foreground objects are crucial for downstream tasks, such as planning. Enhancing our trajectory resampling method requires 4D modeling, which can be challenging and needs further study. We plan to introduce a foreground assets repository, along with agents that control the foreground trajectory, in future work.

---

### Official Review · Reviewer_GaSm · 2024-11-02

**Soundness:** 3
**Presentation:** 3
**Contribution:** 3
**Rating:** 6
**Confidence:** 3

**Summary:**

This paper introduces DOME, a diffusion-based world model for predicting future occupancy frames from past observations. It demonstrates high-fidelity generation, accurately predicting future changes in occupancy space, and can generate long-duration occupancy sequences that are twice as extensive as those produced by previous methods.

**Strengths:**

The model offers two notable strengths: 1) High-Fidelity and Long-Duration Generation: DOME utilizes a spatial-temporal diffusion transformer to effectively leverage historical occupancy data. This architecture captures detailed spatial-temporal dependencies, enabling the generation of high-fidelity predictions across extended timeframes. 2) Fine-Grained Controllability: To tackle the challenge of controlling predictive outputs, DOME incorporates a trajectory resampling method. This enhancement significantly improves the model’s ability to produce controlled and precise predictions, marking a key advancement in diffusion-based modeling. Together, these features establish DOME as a valuable contribution to predictive world modeling, especially for applications that demand both high-detail and controllable long-term forecasts.

**Weaknesses:**

1. In Table 1, could the authors clarify why DOME performs less effectively in the "Others" category? This clarification is important, as it relates to the zero-shot capability of the approach and may highlight limitations in handling diverse or unseen scenarios.
2. Additionally, could the authors provide a comparison of model parameters and computation costs relative to other approaches? This information would help assess DOME's efficiency and scalability in practical applications.

**Questions:**

It would be helpful if the authors provided clear demonstrations with corresponding annotations for each approach. This would enhance the figure’s readability, making it easier for readers to compare and understand the differences between methods.

====================================

Thank you to the authors for the rebuttal. The results look acceptable, so I will keep my score unchanged.

---

> ### Author Response · Authors · 2024-11-21
>
> We thank the reviewer for highlighting the strengths of our model, including its ability to generate high-fidelity, long-duration predictions through spatial-temporal diffusion and its fine-grained controllability via trajectory resampling. We address any further comments or concerns below.
>
> ---
>
> > 1. Q: "In Table 1, could the authors clarify why DOME performs less effectively in the "Others" category? This clarification is important, as it relates to the zero-shot capability of the approach and may highlight limitations in handling diverse or unseen scenarios."
>
> **A:**  We would like to clarify that the lower performance of DOME in the "Others" category is likely due to the inclusion of **small objects**, such as trash bins and birds, in this class. More details about this class can be found in Occ3D [1]. Furthermore, other baseline methods adopt a spatial compression rate of 16, whereas we employed a compression rate of 64. **This higher compression rate** may have adversely affected the representation of small objects, leading to decreased performance in this category.
>
> However, we conducted extensive testing of our model under the same 16 compression ratio settings used by the baseline methods. The results demonstrate that our model outperforms all baseline methods across these settings, further validating its effectiveness.
>
> - [vae_comparison](https://anonymous.4open.science/r/iclr-rebuttal-7BC0/vae_comparsion.jpg)
>
> To further address concerns about generalizability to unseen data, we present the 4D occupancy prediction results of our method on a different dataset (KITTI) in zero-shot setting.
>
> - [kitti_demo](https://anonymous.4open.science/r/iclr-rebuttal-7BC0/kitti.mp4)
>
> [1] Tian, Xiaoyu, et al. "Occ3d: A large-scale 3d occupancy prediction benchmark for autonomous driving." Advances in Neural Information Processing Systems 36 (2024).
>
> ---
>
> > 2. Q: "Could the authors provide a comparison of model parameters and computation costs relative to other approaches? This information would help assess DOME's efficiency and scalability in practical applications."
>
> **A:**  We have now included this information in the table for comparison. As shown in the table, our method achieves **superior performance and demonstrates a shorter inference time** compared to other approaches, with only a slight increase in model parameters. Experiments show that our approach achieves **a great balance between efficiency, scalability, and performance**, highlighting the practicality of DOME for real-world applications.
>
> - [model_comparison](https://anonymous.4open.science/r/iclr-rebuttal-7BC0/parameter_comparison_table.jpg)
>
> ---
>
> > 3. Q:"It would be helpful if the authors provided clear demonstrations with corresponding annotations for each approach. This would enhance the figure’s readability, making it easier for readers to compare and understand the differences between methods."
>
> **A:**  We have reviewed our figures and noted that we did include annotations for method comparisons. Could the reviewer kindly specify which figure they are referring to? This will help us address the issue more effectively and make the necessary improvements promptly.

---

### Official Review · Reviewer_gC2B · 2024-11-03

**Soundness:** 3
**Presentation:** 3
**Contribution:** 3
**Rating:** 6
**Confidence:** 3

**Summary:**

The authors proposed a pipeline to predict future occupancy frames conditioned on historical occupancy observations and trajectory control. The approach entails Occ-VAE and a spatial-temperal diffusion with trajectory encoder and a novel resampling technique.

**Strengths:**

1. The paper is well organized and cleanly presented.
2. Thorough ablations and good experiment results are demonstrated.
3. The trajectory resampling technique for data augmentation is considered novel, allowing for good performance.

**Weaknesses:**

Please see questions.

**Questions:**

1. The authors mentioned around line 123 in the related work section that "BEV methods struggle to convey detailed 3D information due to their top-down projection." However, the authors' approach is still operating in "transform[ed] ... Bird’s Eye View (BEV) style tensor" (line 199). How would the BEV style feature be better than other approaches in maintaining "detailed 3D information" during VAE latent space compression?
2. Section 3.1 mentions the "class embedding" that needs to be learned for OCC-VAE, without much detailed discussion on its purpose and downstream impacts. Would this be, for example, a potential bottolneck for making the model more generalizable?
3. It seems like that, in Tab. 3, the trajectory resampling technique allows for a rather reasonable improvement on the mIoU and IoU metric. Do the authors think of the technique as the central piece of this paper's novelty that allows for SOTA performance? If similar data augmentation is applied to OccWorld or OccLLaMA, would they also perform on par with DOME? (sorry I may not be super familiar with this specific line of work, and I am willing to improve my rating as needed)
4. Missing spaces in line 467 "further demonstrating that DOMEhas strong generalizability" and line 494 ": 4D occupancy forecasting performance.Avg. denotes..." I would suggest further proof reading.
5. Tab. 4 suggests that DOME allows for 32s rollout for long duration generation without explicitly discussing the potential accumulated error. What's preventing the model from generating a even longer rollout if error range is not considered (particularly given the autoregressive approach)?

---

> ### Author Response · Authors · 2024-11-21
>
> We thank the reviewer for recognizing the clarity, organization, and strong experimental results of our work, as well as the novelty of our trajectory resampling technique. We address any questions and concerns below.
>
> ---
>
> > 1. Q: 'How would the BEV style feature be better than other approaches in maintaining "detailed 3D information" during VAE latent space compression?'
>
> **A:** Apologies for the ambiguity in our previous statement. We would like to clarify that **our statement is not contradictory**. First, the statement in our related work **is cited from TPVFormer** [1]. The BEV-style feature in our approach **differs significantly** from the methods referenced in our related work in terms of task. In our method, the BEV-style feature is designed for the task of occupancy compression, whereas those methods are designed for occupancy estimation.
>
> Second, **those methods typically employ pooling operations** (e.g., averaging) along the height dimension to reduce computation costs, which leads to a loss of detailed 3D information. However, this does not significantly affect their task performance. In contrast, **we rearrange the height dimension into the feature channel dimension**, avoiding this information loss and preserving more detailed 3D information.
>
> [1]Huang, Yuanhui, et al. "Tri-perspective view for vision-based 3d semantic occupancy prediction." Proceedings of the IEEE/CVF conference on computer vision and pattern recognition. 2023.
>
> ---
>
> > 2. Q: "What are the purpose and downstream impact of the "class embedding" in Section 3.1? Could it potentially act as a bottleneck, limiting the model's generalizability?"
>
> **A:** The class embedding is **a set of learnable parameters that maps discrete occupancy semantic IDs into embeddings** that can be processed by the encoder.
>
> Regarding generalizability, we want to clarify that **this will not be an issue as long as the occupancy semantic IDs have a unified definition**. However, this could become a limitation when dealing with different semantic IDs, such as in cross-dataset inference. The solution is to simply unify the semantic IDs. We tested this scenario in KITTI dataset, which has entirely different semantic IDs, and it turned out fine.
>
> - [kitti_demo](https://anonymous.4open.science/r/iclr-rebuttal-7BC0/kitti.mp4)
>
> ---
>
> > 3. Q: "Do the authors think of the trajectory resampling as the central piece of this paper's novelty that allows for SOTA performance? If similar data augmentation is applied to OccWorld or OccLLaMA, would they also perform on par with DOME?"
>
> **A:** Our experiments demonstrate that **our novel framework, combined with trajectory resampling**, significantly improves performance. Furthermore, **even without trajectory resampling**, our method outperforms others. As shown in Table 3, Line 3, our model achieves an mIoU of 24.24 and an IoU of 34.28, which are notably higher than those of OccWorld (mIoU: 17.14, IoU: 26.63) and OccLLaMA (mIoU: 19.93, IoU: 29.17).
>
> Additionally, **trajectory resampling can serve as a general data augmentation technique** applicable to other occupancy world models or even LiDAR world models. This is because it alters the data distribution of trajectories without being model-specific
>
> ---
>
> > 4. Q: ' Missing spaces in line 467 "further demonstrating that DOMEhas strong generalizability" and line 494 ": 4D occupancy forecasting performance.Avg. denotes..." I would suggest further proof reading.'
>
> **A:** Thank you for your careful review. We have corrected these typographical errors and conducted a thorough proofreading of the manuscript. The updated version can be seen in the revised paper.
>
> ---
>
> > 5. Q: "Tab. 4 suggests that DOME allows for 32s rollout for long duration generation without explicitly discussing the potential accumulated error. What's preventing the model from generating a even longer rollout if error range is not considered?"
>
> **A:** Thank you for pointing out this concern. We acknowledge that **long-sequence generation is inherently challenging**, even in the well-studied field of video generation [1]. In the context of our task, the challenge of generating a long occupancy sequence arises from two primary factors:
>
> - The limited availability of long ground truth sequences, which in our case are about 20 seconds in duration within the nuScenes dataset.
> - The use of auto-regressive paradigms for generating long occupancy sequences leads to error accumulation, which degrades the overall generation quality.
>
> Compared to baseline methods, which typically generate sequences lasting only 3 seconds, **our method achieves the longest generation duration**.
>
> We also tested longer rollouts and observed issues such as repetitive generation. In future work, we will aim to address this problem, potentially by employing a divide-and-conquer strategy.
>
> [1] Li, Chengxuan, et al. "A survey on long video generation: Challenges, methods, and prospects." arXiv preprint arXiv:2403.16407 (2024).

---

> > ### Comment · Reviewer_gC2B · 2024-11-25
> >
> > I appreciate the authors’ efforts in addressing my question. To further clarify Q5, I was also interested in seeing the error bounds for the demonstrated long rollouts. While the authors make a valid point about the challenges of long-sequence generation, I believe this alone does not sufficiently demonstrate the proposed method’s superiority without quantitative metrics illustrating the quality of the long rollouts. Also, I can't see any revised paper available on OpenReview yet, and the latest version still contains the typos I previously pointed out.

---

> ### Author Response · Authors · 2024-11-26
>
> > 1. Q:"Regarding the quantitative metrics of the long rollout results."
>
> Thank you for your suggestion regarding the testing of quantitative metrics for long rollouts. We have tested our FVD metric following the approach in VideoGPT [1], and the results are listed below. We observed that our method outperforms the baseline.
>
> | Method      | FVD↓  |
> |-------------|-------|
> | OccWorld    | 0.844 |
> | DOME (ours) | 0.349 |
>
> In future work, we will explore different paradigms for generating long sequences, such as first generating key frames and then interpolating for longer sequences. Additionally, we aim to explore more reasonable quantitative metrics for evaluating the generation of long occupancy sequences.
>
> [1] Yan, Wilson, et al. "Videogpt: Video generation using vq-vae and transformers." arXiv preprint arXiv:2104.10157 (2021).
>
> ---
> > 2. Q:"Regarding the lack of updates on the revised paper."
>
> We apologize for forgetting to update the revised paper. we have now updated the PDF.

---

### Meta-Review · Area_Chair_41CK · 2024-12-22

**Metareview:**

This paper introduces DOME, a diffusion-based occupancy world model for predicting future occupancy frames conditioned on historical observations and trajectory control. The approach leverages a spatial-temporal diffusion Transformer for long-duration generation and a trajectory resampling method for enhanced controllability. DOME achieves SOTA performance on the nuScenes dataset, demonstrating superior results in occupancy reconstruction and long-term forecasting. After careful discussion in the rebuttal period, the reviews remain divided. One major concern is trajectory resampling and the removal of foreground objects, mentioned by reviewer JrJ2. Indeed, directly removing dynamic objects is not practical as it requires additional ground-truth labels. Also, dynamic objects are critical in autonomous driving, especially for downstream object detection. Therefore, removing them, despite improving occupancy prediction performance, might lead to degenerated perception and even harm end-to-end performance. Given this critical concern, AC recommends rejecting this paper.

**Additional Comments On Reviewer Discussion:**

Most concerns regarding technical details and additional experiments addressed in the rebuttal process. However, as mentioned above, one critical concern (by JrJ2), i.e. the removal of dynamic objects, is not adequately addressed in the rebuttal. Also, in addition to occupancy prediction, authors should provide additional downstream task performance to justify the effectiveness of the proposed world model.

---

### Decision · Program_Chairs · 2025-01-22

Reject